ecology, ecosystems, environmental science

feedbacks, grand challenges, biodiversity, science–policy, ecosystem functioning, socioecological systems

**Author for correspondence:**
Mary I. O'Connor
e-mail: oconnor@zoology.ubc.ca

# Grand challenges in biodiversity–ecosystem functioning research in the era of science–policy platforms require explicit consideration of feedbacks

Mary I. O'Connor[1,2], Akira S. Mori[4], Andrew Gonzalez[5], Laura E. Dee[6], Michel Loreau[7], Meghan Avolio[8], Jarrett E. K. Byrnes[9], William Cheung[2,3], Jane Cowles[10], Adam T. Clark[11], Yann Hautier[12], Andrew Hector[13], Kimberly Komatsu[14], Tim Newbold[15], Charlotte L. Outhwaite[15], Peter B. Reich[16,17,18,19], Eric Seabloom[10], Laura Williams[16], Alexandra Wright[20] and Forest Isbell[10]

[1]Department of Zoology, [2]Biodiversity Research Centre, and [3]Institute for the Oceans and Fisheries, University of British Columbia, Vancouver, Canada
[4]Graduate School of Environment and Information Sciences, Yokohama National University, Yokohama, Japan
[5]Department of Biology, McGill University, Montreal, QC, Canada
[6]Department of Ecology and Evolutionary Biology, University of Colorado at Boulder, Boulder, CO, USA
[7]Theoretical and Empirical Ecology Station, CNRS, Moulis, France
[8]Department of Earth and Planetary Sciences, Johns Hopkins University, Baltimore, MD 21218, USA
[9]College of Science and Mathematics, University of Massachusetts–Boston, Boston, MA, USA
[10]Department of Ecology, Evolution and Behavior, University of Minnesota, St Paul, MN, USA
[11]Institute of Biology, University of Graz, Holteigasse 6, 8010 Graz, Austria
[12]Ecology and Biodiversity Group, Department of Biology, Utrecht University, Padualaan 8, 3584 CH Utrecht, The Netherlands
[13]Department of Plant Sciences, University of Oxford, Oxford, UK
[14]Smithsonian Environmental Research Center, Edgewater, MD, USA
[15]Centre for Biodiversity and Environment Research, Department of Genetics, Evolution and Environment, University College London, London, UK
[16]Department of Forest Resources, University of Minnesota, St Paul, MN 55108 USA
[17]Hawkesbury Institute for the Environment, Western Sydney University, Penrith, NSW 2753, Australia
[18]Institute for Global Change Biology, and [19]School for Environment and Sustainability, University of Michigan, Ann Arbor, MI 48109, USA
[20]Biological Sciences Department, California State University Los Angeles, 5151 State University Drive, Los Angeles, CA, USA

MIO, 0000-0001-9583-1592; ASM, 0000-0002-8422-1198; LED, 0000-0003-0471-1371; AH, 0000-0002-1309-7716; KK, 0000-0001-7056-4547; TN, 0000-0001-7361-0051

Feedbacks are an essential feature of resilient socio-economic systems, yet the feedbacks between biodiversity, ecosystem services and human well-being are not fully accounted for in global policy efforts that consider future scenarios for human activities and their consequences for nature. Failure to integrate feedbacks in our knowledge frameworks exacerbates uncertainty in future projections and potentially prevents us from realizing the full benefits of actions we can take to enhance sustainability. We identify six scientific research challenges that, if addressed, could allow future policy, conservation and monitoring efforts to quantitatively account for ecosystem and societal consequences of biodiversity change. Placing feedbacks prominently in our frameworks would lead to (i) coordinated observation of biodiversity change, ecosystem functions and human actions, (ii) joint experiment and observation programmes, (iii) more effective use of emerging technologies in biodiversity science and policy, and (iv) a more inclusive and integrated global community of biodiversity observers. To meet these

challenges, we outline a five-point action plan for collaboration and connection among scientists and policymakers that emphasizes diversity, inclusion and open access. Efforts to protect biodiversity require the best possible scientific understanding of human activities, biodiversity trends, ecosystem functions and—critically—the feedbacks among them.

## 1. Dynamic feedbacks are causes and consequences of biodiversity change

Increasing recognition of irreversible biodiversity change and unsustainable ecosystem exploitation has spurred unprecedented collaboration among scientists and policymakers worldwide to mitigate these ecological crises [1–5]. Biodiversity is in crisis as a result of habitat loss, overharvesting and other pressures associated with humanity's accelerated use of natural resources. The diversity of life—from genes to social–ecological systems—plays a major role as both a driver of ecosystem dynamics throughout the biosphere and a response to changes in ecosystem processes; greater biodiversity can enhance ecosystem functioning [6–8] and 'nature's contributions to people' (see Glossary in box 1), while also responding to human activities such as cultivation or harvesting. Biodiversity, its responses to human activities, and the benefits it can provide to human wellbeing are now at the centre of global science–policy initiatives such as the Intergovernmental Panel on Biodiversity and Ecosystem Services (IPBES) and the new Global Biodiversity Framework of the Convention on Biological Diversity (CBD) [2].

The science underpinning these major initiatives has clearly demonstrated direct effects of biodiversity on ecosystem functioning and human wellbeing (B-E-H) (figure 1), as well as dynamic feedbacks (§2) that influence how B-E-H system components change over time. Direct effects include the positive effect of species diversity on productivity and nutrient dynamics in plant and animal systems [14,15], increased productivity and food quality benefitting humans through an ecosystem service such as food provision [7,16–18], and food management systems that facilitate biodiversity [19,20] (figure 1). Direct effects also include the human actors benefiting from nature, while also engaging in activities that benefit or harm biodiversity. Direct effects alone cannot tell the full story [21]; system dynamics commonly feature feedbacks (figures 1 and 2), and the biosphere is a system comprising the diversity of life on earth, ecosystems and human built structures and systems.

The next generation of biodiversity scholarship will more effectively understand feedbacks as essential features of any focus on biodiversity and how it changes in relation to human activities and ecosystem functioning [22]. This knowledge will better inform policy platforms and actions taken in compliance such as monitoring biodiversity. Here, we consider biodiversity, ecosystem functioning and humanity as components of a system, and in doing so, we highlight the central role that feedbacks play in sustaining dynamic relationships among these components (§2). Next, we briefly review how current leading policy platforms consider the role of feedbacks and highlight opportunities for strengthening consideration of feedbacks (§3). We then identify key scientific knowledge gaps (§4)

that we suggest limit the full uptake of scientific understanding into policy platforms and deserve organized and collaborative investment for rapid progress. Finally, we outline an agenda for collaborative action (§5) to meet these challenges to support policy-relevant science in a changing world, as our understanding of that world also changes.

## 2. Feedbacks are essential features of biodiversity–ecosystem functioning–human relationships

Biodiversity and its relationships to ecosystem functioning and human wellbeing depend on feedbacks within and between these system components (figures 1 and 2) [23–25]. The feedback concept is often used to describe specific dynamic interactions that are considered real and observable in human ecological systems. The feedback concept is used to refer to interaction networks [26] or dynamics of a complex system that amplify or dampen an outside signal or effect. The concept can be used more loosely as a communication tool, for example, when a species' 'final descent into extinction' reflects synergistic effects of multiple stressors, the synergy may be referred to as involving a feedback [27]. Feedbacks between biotic and abiotic processes driving the global carbon cycle have received great attention in climate science and policy because they cause systems to change in non-intuitive ways over time [23,25]. Additionally, feedbacks between human and ecological subsystems have become an important area of interdisciplinary research and for guiding discourse [28–30]. These research programmes all contribute to the solution we are addressing here—to better understand feedbacks specifically in the B-E-H system as a whole [31], and how best to apply this understanding to broad scale policy, communication and knowledge integration programmes.

A simple definition of feedback is when one part of a system affects another part of that system that in turn affects the first part; in other words, a system output affects the input of the same system. This definition is consistent with systems biology, recognizing feedback as a control mechanism in complex systems. *Positive feedbacks* are self-reinforcing, and can drive rapid change and even destabilize systems [32] (figure 2*a*). *Negative feedbacks* (figure 2*b*) are self-dampening and stabilizing, and can buffer systems against change [33,34]. Modelling feedbacks as opposed to direct effects involves approaches such as equations that relate the *behaviour* over time of a system to the *state* of that same system in some way. It is this self-dependent relationship that distinguishes models with dynamic feedbacks from models that include direct and indirect effects but do not relate these in feedbacks (figure 2).

Feedbacks explain change and stability in systems involving biodiversity, ecosystem functioning and human wellbeing. Among the processes that maintain biodiversity, feedbacks determine stability and future trajectories of population, community and ecosystem dynamics [33,35,36], from shallow lakes [37] to tropical rainforests [38] to coral reefs [39]. First-order biological processes—growth and reproduction—are positive feedbacks [40]. One of the most pervasive feedbacks in ecological systems is density dependence of population dynamics, in which population density at one time influences population growth at a future time, which in turn influences future population density (figure 2). Stronger

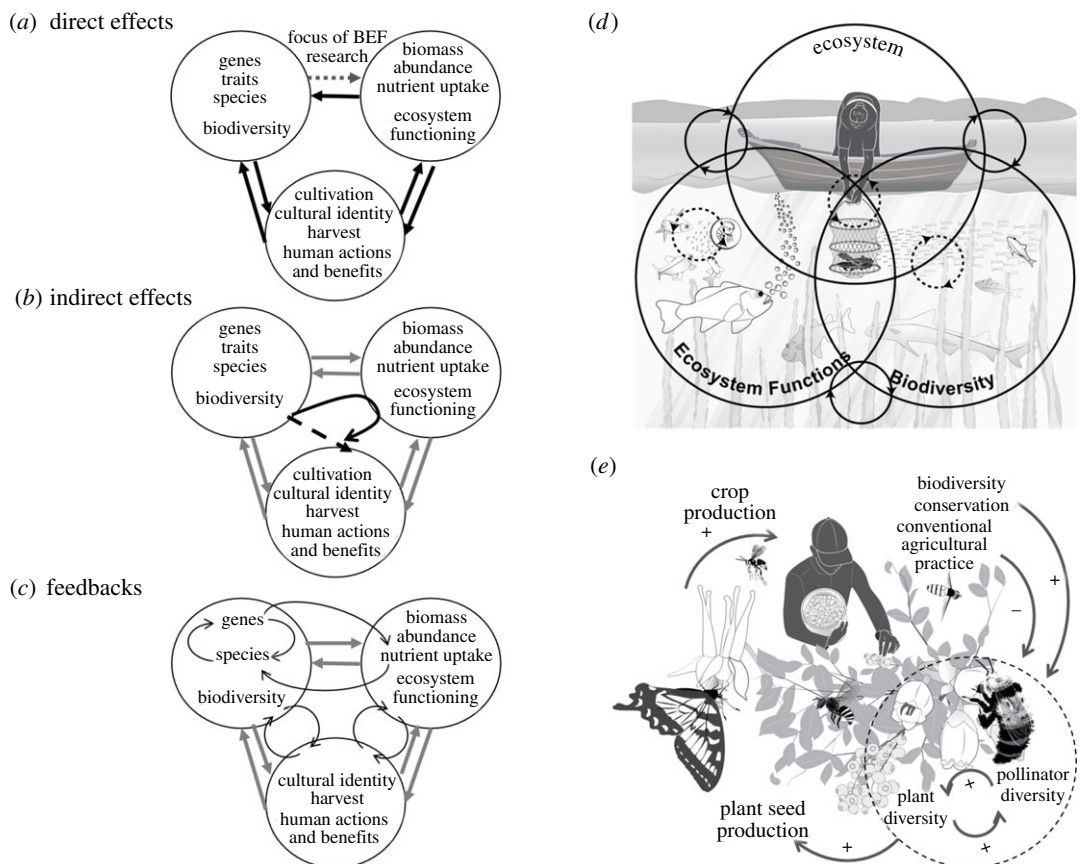

**Figure 1.** Direct effects, indirect effects and feedbacks in the biodiversity–ecosystem functioning–human wellbeing (B-E-H) system. (*a*) Direct effects are one-way effects of, for example, species richness on an ecosystem function; biodiversity–ecosystem functioning (BEF) research has emphasized the direct effect of diversity on functioning (dashed arrow). (*b*) Indirect effects are summed direct effects. (*c*) Feedbacks are iterative and ongoing, often looping, effects of system components on each other. (*d*) In an aquatic example, invertebrate and vertebrate diversity enhance ecosystem functions such as biomass production that may be harvested for food and livelihood by people. Harvesting may maintain some fish at high population growth rates by reducing population densities thereby maintaining biodiversity; (*e*) in an agricultural plant–pollinator system, a full feedback between diversity, plant seed production and human activities has led to recognition that conservation measures to protect pollinator diversity may benefit humans by enhancing crop yields.

---

**Box 1.** Glossary

**Biodiversity:** variety of life. We use the concept to include people in the living earth system; biodiversity is measured at many scales and in many ways, from genetic diversity to functional diversity to behavioural or cultural diversity.

**Feedback:** modification or control of a process by the results or effects of the same process.

**Ecosystems:** joint biotic/abiotic systems of life, characterized by dynamic stocks and fluxes of energy, materials and information and their feedbacks.

**Biodiversity–ecosystem functioning (BEF) relationships:** refers to the relationship between diversity *per se* and the magnitude and stability of an ecosystem functions. BEF refers to the role diversity plays in ecosystem functioning that is over and above the importance of total abundance, biomass or composition of the biological assemblage [9].

**Ecosystem functioning:** the processes of energy flow (e.g. primary production), material cycling (e.g. carbon cycling) and information processing (e.g. evolution) carried out by living systems. Functions are understood to reflect interaction networks involving multiple genetic and functional elements of biodiversity, and include stocks and pools of biomass, elements and energy forms.

**Ecosystem services:** the value of ecosystem functions to people [10], and originally, defined as ecosystem-based goods and services for human wellbeing. Although different opinions exist such as that ecosystem services could be viewed as 'rights-based approaches to biodiversity conservation and sustainable use' [11], it is important to emphasize that the value can be assessed in a variety of ways, from economic values to cultural values, in intrinsic, instrumental or relational systems [12,13].

**Nature's contributions to people (NCP):** a pluralistic view for the value of ecosystems and ecosystem functioning to people [12,13]. Peterson *et al*. [13] expect the view to encourage a recognition of pluralism and the need for a richer process of articulation, translation and discussion among many different perspectives on people's relationship with nature.

---

density dependence *within* species than *among* species is one of the primary explanations for the persistence of biodiversity in nature and for the positive relationship between biodiversity

and ecosystem services [40–42] (figure 2*c*). Negative (dampening) density-dependent feedbacks of predation, disease and pathogens on species' performance cause diverse systems to

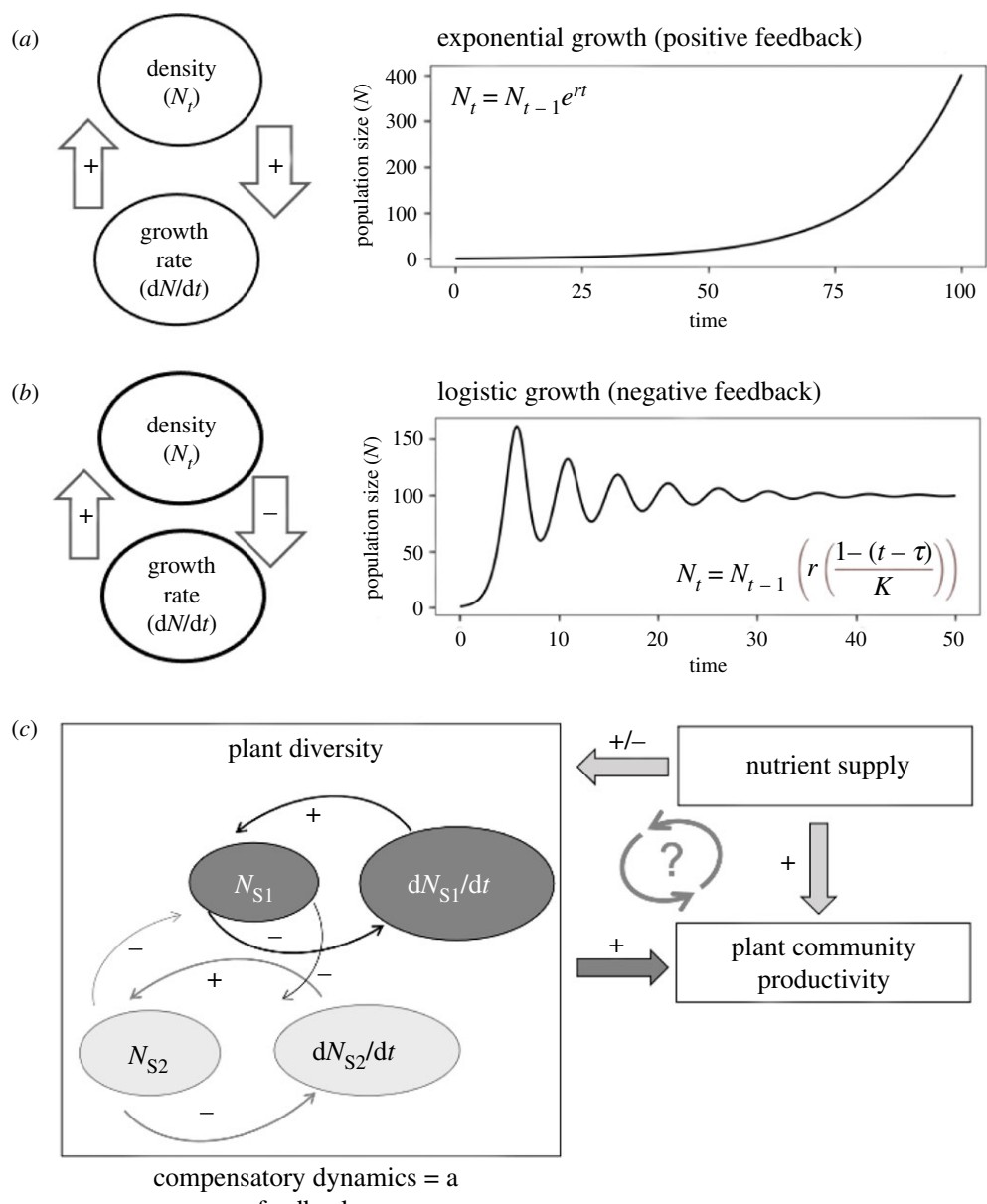

**Figure 2.** Feedbacks in (*a,b*) population dynamics and (*c*) community dynamics. (*a*) Positive and (*b*) negative feedback between population growth rate (d$N$/d$t$) and population density ($N_t$) in closed systems comprising one population. (*c*) Density-dependent feedbacks among plant populations and species can lead to a positive effect of plant diversity on plant productivity (an ecosystem function). Nutrient supply can modify the relationship between diversity and productivity by directly enhancing productivity and by changing plant diversity and composition. Whether there is a feedback between nutrient supply, diversity and productivity is not yet fully resolved (the grey question mark).

maintain diversity and ecosystem functions over time more than less diverse systems [24,43,44] (though these ecological interactions can also be involved in positive feedbacks). Density-dependent processes are at the heart of compensatory dynamics in which a decline in density of a competitive dominant allows competitors to increase in abundance and maintain ecosystem functions in a negative feedback [40,42,45]. In some cases, we can study the dynamics of part of the system—for example, we isolate feedbacks that maintain diversity when we study compensatory dynamics—but to fully understand the problems we now face, we have to continue the research process by expanding our focus from the dynamics of a subsystem to the more complex B-E-H systems.

There are many examples of change that we now understand to depend on feedbacks between biodiversity, ecosystem processes and human activities (e.g. [24,33,35,46]). Feedbacks in the pollinator/plant system provide a particularly good example [47,48] (figure 1e). Pollinator functional diversity can

increase pollination and plant seed production [49,50], and plant diversity through niche complementarity (different pollinators pollinate different plants) as well as changes in pollinator behaviour [51,52]. This creates a positive feedback: pollinator diversity affects plant diversity which can in turn feedback to enhance and sustain pollinator diversity (figure 1e). Further, humans benefit when the plants are of cultural or agricultural value. Some agricultural practices, land use change and pollution have dramatically reduced pollinator abundance and diversity [53,54], potentially contributing to loss of value in crop yields. Negative effects of human activities on pollinator diversity and the recognition of the feedback of human activities to human benefits through crop pollination have motivated conservation and management actions that focus not only on reducing pollution but also on restoring diversity in plant–pollinator–human systems [55]. The inclusion of conservation activities focused on pollinator diversity creates a feedback involving humans, pollinators and plant diversity (figure 1e).

## 3. Feedbacks have been under-emphasized in major science-based policy platforms

Major science-based policy platforms guide decisions about a broad range of actions that impact biodiversity change, including setting targets for sustainability (UN Sustainable Development Goals, SDGs) and the targets in the post-2020 Global Biodiversity Framework of the CBD [56]. The IPBES framework [1] provides the broader community with a system for understanding how biodiversity, inclusive of humanity and human diversity (box 1), is related to a sustainable biosphere [57]. This framework is offered with the purpose of aligning assessments of change and knowledge development in biological and social sciences with the policy challenges of the coming decades [11,57]. These challenges include state-level investments in biodiversity observation and conservation [56,58], as well as integration of policies to jointly mitigate climate change and biodiversity change [3,59,60], and to manage food systems for nature-positive outcomes and sustainable food provision [61].

The IPBES platform also channels and motivates scholarship and scientific research to fill gaps and improve methods for modelling scenarios. It relies on synthesis of scientific evidence for the causes and consequences of biodiversity change, combined with scientific models to project future scenarios [62]. There is little mention of full feedback cycles between biodiversity and ecosystem functioning (e.g. figure 1*a*) in the summary of models used to generate projections and scenarios for the most recent IPBES report. The few examples are in the integrated assessment models involving social and economic systems coupled with natural systems [62]. The assessment report indicates that feedbacks are identified as an *outcome* of integrated system models, rather than an architectural feature [62]. The IPBES approach to scenarios does include qualitative modelling methods that can capture feedbacks, though these methods are largely restricted to smaller-scale social–ecological system studies as in fisheries (e.g. [63]), yet a major gap exists in the integration between different types of interactions in order to more comprehensively characterize the major feedbacks between or within, for example, ecosystems and fisheries. The IPBES methods assessment report notes that 'Failure to consider such [feedback] dynamics can potentially render scenario analysis incomplete, inconsistent or inaccurate'. IBPES authors and ecosystem modellers also highlight the risks associated with including feedbacks based on wrong or incomplete understanding. It is recognized that knowledge gaps—both scientific and in the general understanding and application of science—are a barrier. As we move to consider feedbacks more, it is important to recognize that there are many ways to do this, including quantitative modelling and heuristic consideration as illustrated in the pollinator example (figure 1*e*).

## 4. Key knowledge gaps that present grand challenges for biodiversity research

Our survey revealed seven knowledge gaps in biodiversity science when we considered the B-E-H system as a whole system, rather than take previously prevalent perspectives that emphasize two of the three components—BEF that tends to consider human activities as outside the system, or

socioecological systems (SES) in which biodiversity and functioning are lumped into one component. Filling these knowledge gaps requires targeting feedbacks as scientific research subjects, and considering how assessments and policies can best reflect this knowledge development. We suggest that these challenges might be used to prioritize major investment to expand the BEF paradigm and enhance our knowledge frameworks to support biodiversity policies and to realize sustainability goals (see §5).

(1) *We cannot robustly relate current or recent temporal trends in biodiversity to likely future trajectories of biodiversity change in most cases.* As we have noted above, future biodiversity, and diversity's contribution to ecosystem services, may not be accurately projected by extrapolating a historical trend in biodiversity forward in time because of feedbacks among biodiversity, ecosystem function and human activities [21,36,62,64]. Consideration of feedbacks highlights that human activities and ecosystem functioning are part of changing biodiversity in the system, and forces us to reframe this question such that we cannot only examine biodiversity trajectories. To estimate long-term behaviour of a B-E-H system in scenarios that might be used to guide decisions, the dynamics—and in particular, feedbacks such as how biodiversity change and its causes can influence future biodiversity—need to be considered.

(2) *We do not understand the B-E-H system well enough to relate observed recent trends in biodiversity to likely future trends in ecosystem function and human wellbeing.* Dynamics of one part of the system (for example, diversity) depend on other parts of the system (humans, ecosystem functions), and vice versa. Because feedbacks characterize how biodiversity, ecosystem functioning and human activities change together over time, projected future trajectories or scenarios of diversity, ecosystem functioning or human wellbeing require consideration of all three components. One pervasive consequence of this knowledge gap is the persistent decoupling of biodiversity and functioning in assessment and monitoring programmes; most of the biodiversity observations being assembled for biodiversity change assessments (e.g. BioTIME, PREDICTS, GEO BON) do not systematically include accompanying measures of ecosystem processes or human activities. Though GEO BON is moving in this direction with essential ecosystem variables, such an advance must be made in the context of statistical approaches that can allow detection and attribution of joint changes in biodiversity, ecosystem functioning and human wellbeing.

(3) *Trends in B-E-H components depend on scale, yet we still do not understand exactly how, and what feedbacks play in determining scale dependence.* Trends observed at one scale do not necessarily predict trends at higher or lower spatial resolutions [65], and this gap is a major barrier to synthesizing observations across studies and programmes to infer biodiversity change [22]. We require new theory to guide experimental tests and observation programmes that allow us to more deeply understand feedbacks between diversity change and ecosystem functioning, and how these are linked in coupled human–natural systems across scales of space, time and organization [22] (figure 3). Such theory and experimental work would be explicit about temporal patterns in BEF, spatial and temporal variation, and would identify links between feedbacks

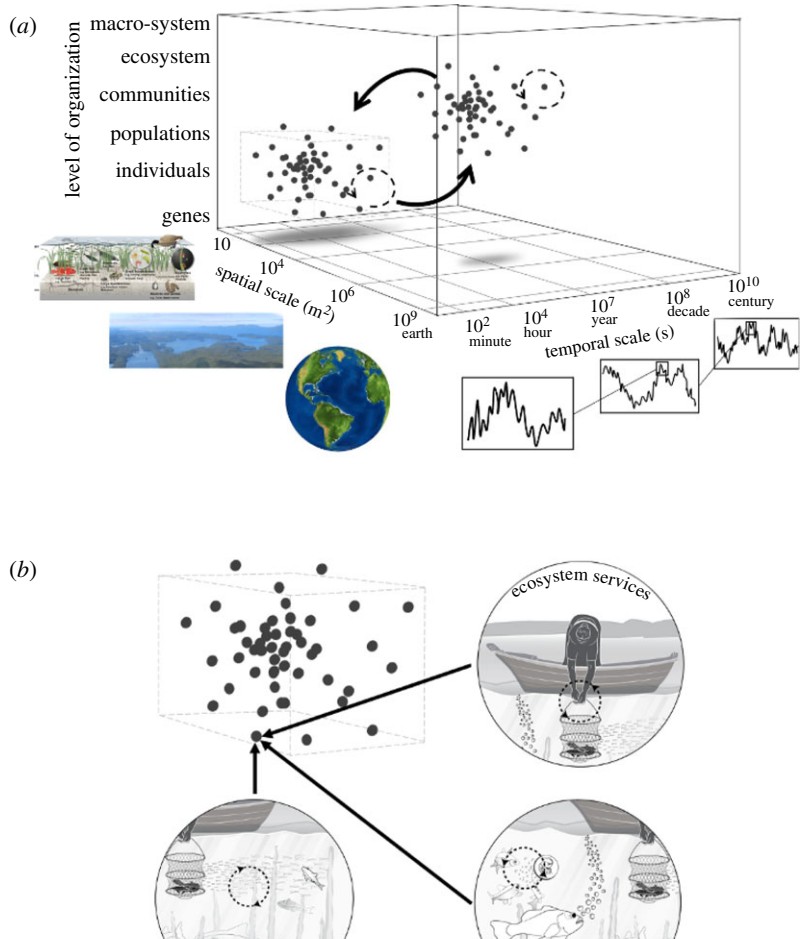

**Figure 3.** Models, experiments and observation systems are needed that explicitly address feedbacks and scales of space, time and biological organization. (*a*) Many programmes tend to focus in one part of this space—for example, generating data within the dashed box—and we argue for approaches that relate observations at multiple (modified from Gonzalez *et al.* [22]). (*b*) Hypothetical data copied from (*a*), illustrating that we should strive for observations and understanding of how biodiversity, human activities and ecosystem functions change at the same levels of spatial and temporal resolution, in the context of other spatial and temporal processes (*a*). (Online version in colour.)

involving ecosystem functioning and multiple dimensions of diversity, and the role that human systems play in these biodiversity–ecosystem functioning linkages.

(4) *Experimental tests for direct BEF effects have omitted feedbacks.* The majority of experimental tests of the relationship between BEF conducted in the last two decades has employed an experimental design that intentionally disrupts potential feedbacks—for example, by weeding out species that colonize [66] or by replacing species that are lost [67] over the course of the experiment to maintain diversity treatments. Though this approach clearly isolates direct effects of biodiversity on ecosystem functions (figure 1*a*), in doing so these procedures prevent feedbacks (e.g. figure 2*c*) from playing out over time. Consequently, hundreds of experiments frequently reviewed and synthesized as strong evidence for direct effects of diversity on ecosystem functioning [6,8] (figure 1*a*) cannot be used to demonstrate consequences of the feedbacks between diversity and functioning because each system studied was controlled to prevent them from occurring.

(5) *Human–biodiversity feedbacks are still not well understood,* allowing to persist a perception within the western science framing that people affect biodiversity but that there is little feedback from biodiversity to people [1,29,31,61,68].

The current IPBES framework acknowledges this knowledge gap: a high-level message (Key Finding 3.3) is that scenarios and models 'need to be better linked in order to improve understanding and explanation of important relationships and feedbacks between components of coupled social–ecological systems' [62]. The challenge we face is, therefore, to integrate the multiple human (behavioural, demographic, social, cultural, political, economic, institutional) components of the B-E-H system in ways that reflect the dependence of human wellbeing on biodiversity as well as the effects of humans on biodiversity [29,69]. Meeting this challenge requires transdisciplinary scholarship to identify the dominant feedbacks and feedbacks of particular interest to stakeholders, as well as to develop approaches to model these feedbacks and to communicate their effects on system projections and scenarios.

(6) *Develop an operational understanding of how different dimensions of biodiversity are involved in feedbacks over time.* Estimates of biodiversity change are based on observations of some dimension of biodiversity as defined in conventional scientific concepts: alleles, genes, traits, species (or operational taxonomic units, OTU) and models of phylogenies. Not only do we still require great investment in organized biodiversity sampling and monitoring [9,70],

we lack scientific knowledge to relate changes in observed diversity at different levels of biological organization (genes versus species; figure 3) to changes in diversity at other levels, changes in ecosystem functioning and feedbacks between them. One key element of BEF feedbacks is trait expression, linking information in genes and genomes to development and phenotypic variation, and as such BEF feedbacks influence which genes and genomes persist in communities [71]. We require new theory, models and empirical understanding to relate trait expression to underlying genetic diversity, and to explain variation in patterns of trait expression in space and time as they relate to ecosystem functioning and human actions.

(7) *Develop theory and workflows that explicitly relate information from emerging technologies to knowledge that can be used to deepen our understanding of feedbacks.* Technological tools for observing biodiversity allow high throughput and remote sensing of biodiversity at the finest levels of biological organization (viruses, genes, microbes) as well as some measures of ecosystem functions [72–74]. As vast amounts of observational data become available, we face the challenge of understanding how to interpret them in the context of dynamic feedbacks. Feedbacks are difficult to detect from most observational datasets because they require coordinated observations of several facets of a system (e.g. biodiversity, an ecosystem function such as biomass production, human use of the biomass, plus any human–biodiversity interactions), and in nearly all cases, these coupled measurements are not made. Many observations of biodiversity cannot be robustly integrated into models of change over time without accompanying theory and empirical evidence for relationships between observations and the system components they represent.

## 5. Agenda for action

We have outlined seven gaps in B-E-H scientific knowledge that limit our current capacity to assess changes to the biosphere. Resolving these knowledge gaps will require investment in scientific research programmes worldwide to employ diverse, interdisciplinary and even transdisciplinary approaches in the field, in the lab and *in silico*. Outside specialist research communities, B-E-H feedbacks and their consequences are not well represented in conceptual diagrams and models used by policy experts and decision makers to understand biodiversity change and its likely consequences over time. Greater emphasis on this representation can help minimize overlooking this important concept when identifying priorities for biodiversity observation or multifaceted conservation opportunities. Further, many knowledge systems beyond science—such as traditional knowledge systems—include knowledge of feedbacks [29,69,75], and therefore an emphasis on feedbacks may provide another scaffold to integrate biodiversity understanding across diverse forms of knowledge. Here, we outline five 'action items' for implementing the research agenda to maximize benefits to the science–policy community. This agenda is intended to guide knowledge production, but does not outline the full process of informing policy; that important process needs additional consideration beyond the scope of this article.

(1) *Collaborate and connect.* We must convene and support collaborations and knowledge development that reflects the ways people know and interact with biodiversity. The action required is to come together to identify knowledge development priorities at local, regional and global scales that reflect the depth and diversity of how humans and biodiversity are co-dependent. We must take the time to listen and learn from each other, and build from these conversations to the observation and solutions programmes we call for. Doing so will result in an inclusion of a broader range of knowledge systems and perceptions of human–biodiversity interactions [76], benefitting an understanding of feedbacks that is both globally and locally relevant worldwide. People serving as observers, knowledge keepers and knowledge users, as ecosystem service beneficiaries and decision makers, play critical roles in the actual B-E-H feedback cycles, because assessment and management are part of the cycles. Scientific and science–policy collaborations in biodiversity research should strive for cultural, geographical, political and ethnic diversity among researchers and within research projects [76]. We can build on existing science–community partnerships and extending these into biodiversity observation and assessment networks [77].

(2) *Build and sustain multi-scale models to develop and revise scenarios of biosphere change.* Though models exist to produce biodiversity scenarios for the future [40,78], we must double down on our capital and personnel investments in these models to not only simulate changes in biodiversity but also the feedbacks between biodiversity change and changes in human activities and ecosystem functions. To serve the needs of science and society, we must be able to update these models as new observations become available, and to produce scenarios at a range of scales relevant to human decisions—from the scale of a plot of land to that of a country or the globe. Further, we must be modelling biodiversity in the context of the full system, which may be achieved by integrating biodiversity models with other models such as climate models or integrated assessment models [5,79]. These models must be developed and improved in conjunction with the increased effort in biodiversity observatories, advancing statistical procedures for robustly detecting and attributing change, and within the context of the kinds of decisions that will need to be made.

(3) *Build and sustain national and global observatories for temporal change in biodiversity, ecosystem functioning and human activities.* Integrated observations must be made at different spatial scales with worldwide coverage [72], going beyond the *ad hoc* approaches to sampling of biodiversity that has produced a set of observations that is highly biased to developed countries and terrestrial habitats [22,40,80]. To meet the challenges we outline above, observation programmes based on international collaborations and local investment must jointly and simultaneously observe biodiversity change, ecosystem functioning change and human activities—such an integrated global biodiversity observation system goes beyond existing infrastructure for most places [58,70]. Further, biodiversity change observatories need to be comprehensive in their inclusion of areas and biomes on our planet, breaking the historical pattern of emphasis on developed countries and the socially dominant

communities within them [58,81]. New approaches, such as that proposed by Kühl *et al.* [77], must emphasize community involvement and data collection supported by and integrated within a broader context of biodiversity assessment. To succeed, these require investment and action as called for here and by others [77,81,82].

(4) *Experimentally and iteratively test the models and re-evaluate our understanding.* To understand feedbacks, observational programmes (Action 3) should be guided by theory that includes feedbacks, and coupled with experimental programmes to understand feedbacks. As with observatories, the experimental and modelling programmes must be run by collaborations of scientists, modellers and end users from a broad range of biomes, countries and cultural backgrounds, specifically including indigenous and local peoples from the global north and south. This action item is to increase investment in experimental programmes that help to fill specific gaps in our understanding of biodiversity change, and to prioritize those programmes led by multi-sector and multi-disciplinary research and data user teams.

(5) *Identify and support a sustained organizational structure.* A leadership team must assemble, must be able to draw on existing scientific knowledge and work with the research community to develop research programmes. The leadership team must facilitate diversity and comprehensive inclusion of nature and people in the research programmes and associated policy development programmes, can promote the research agenda to potential users and supporters, can lead public engagement activities, and can ensure fully open science practices and data archiving so the findings are available to everyone in the world. The structure of the leadership team should be consistent with current values, and consider collaborative networks and other social structures in its design.

Along the way, the research community will need to confront additional logistical challenges that currently limit rapid scientific advances. These have received attention elsewhere, and resolving these challenges is critical the success of the agenda we have outlined here. These include (i) the current lack of open science and the fact that data for BEF from many places is not curated or made available in a central database [82], (ii) limited technology integration such that observations from different methods are not spatially coordinated [58] and (iii) the clear need for more balanced engagement from the global community [76].

## 6. Conclusion

Feedbacks between human wellbeing and BEF have been appreciated and understood for millennia. Yet, only in recent decades has scientific progress led to recognition of the importance of feedbacks among biodiversity, functioning and people across scales. Despite this recognition, and major progress with models, experiments and observations, major challenges remain to integrate this knowledge with new capabilities to meet the policy challenges of the coming decades. As major policy-guiding scientific assessments grow in importance, it is essential to keep striving for the scientific advances, and in particular theoretical advances, that will foster integration of state-of-the-art scientific understanding with international and local policy objectives. There is no substitute for knowledge of feedbacks. The effects of feedbacks over time cannot be approximated by static representations of direct effects [21]. Many authors have noted that without a fuller scientific understanding of feedbacks that link biodiversity change, ecosystem functioning and human wellbeing, we risk making decisions based on modelled futures that do not capture the full range of likely possibilities [26,64,69]. We cannot afford this just when we need science urgently to guide our planning for the future. By investing in science and supporting collaborative and interdisciplinary partnerships [83] we can realize the fullest potential of a collective knowledge system to project possible futures and act on our understanding of those projects in the best possible way for our planet.

Data accessibility. This article has no additional data.

Authors' contributions. Manuscript original draft and outline: M.I.O., A.G., M.L., A.S.M., L.E.D., F.I., J.C. Manuscript editing: all authors. Figure design and production: M.I.O., C.L.O., A.H., F.I., L.E.D., A.G. Content contribution: all authors.

All authors gave final approval for publication and agreed to be held accountable for the work performed therein.

Competing interests. We declare we have no competing interests.

Funding. This research was supported by the US National Science Foundation (NSF), Long-Term Ecological Research (LTER) grant nos DEB-0620652, DEB-1234162 and DEB-1831944, LTER Network Communications Office grant no DEB-1545288, Long-Term Research in Environmental Biology (LTREB) grant nos DEB-1242531 and DEB-1753859 and Biological Integration Institutes grant no. NSF-DBI-2021898. M.L. was supported by the TULIP Laboratory of Excellence (grant no. ANR-10-LABX-41). A.G. is supported by the Liber Ero Chair in Biodiversity Conservation. F.I. is supported by a NSF CAREER award (DEB-1845334).

Acknowledgements. We are grateful to Dr Julián Idrobo and Dr Matthew Whalen for thoughtful and constructive feedback on a draft of this manuscript.

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
