## [Peer Review File · Proceedings of the Royal Society B: Biological Sciences]

Review History

RSPB-2021-0783.R0 (Original submission)

Review form: Reviewer 1

Recommendation

Major revision is needed (please make suggestions in comments)

Scientific importance: Is the manuscript an original and important contribution to its field?

Excellent

General interest: Is the paper of sufficient general interest?

Excellent

Quality of the paper: Is the overall quality of the paper suitable?

Good

Is the length of the paper justified?

Yes

Should the paper be seen by a specialist statistical reviewer?

No

Do you have any concerns about statistical analyses in this paper? If so, please specify them explicitly in your report.

No

It is a condition of publication that authors make their supporting data, code and materials available - either as supplementary material or hosted in an external repository. Please rate, if applicable, the supporting data on the following criteria.

Is it accessible?

N/A

Is it clear?

N/A

Is it adequate?

N/A

Do you have any ethical concerns with this paper?

No

Comments to the Author

I enjoyed reading this ms and I think it makes an important contribution, setting out clearly why feedbacks are important in B-E-H research, how they've been overlooked, and what could (and should) be done about this. Some parts are stronger than others - I think it builds really well to the end of Section 4, but then peters out a little and the final couple of sections are big on grand sweeping statements but lacking a little in specifics. However overall this is well conceived and contains some excellent points. Additional comments follow, more or less in the order in which they arise in the ms.

Section 1 convinces me of the importance of feedbacks and sets out the structure of the piece really clearly. However, I don't think it is very well supported by fig 1, which in my view could do a much better job of explicitly setting out the feedbacks. As it stands I found it rather difficult to follow, and to identify where feedbacks fit, what exactly they entail, and how they have typically been missed in B-E-H work. 1A could be better annotated I think to specifically identify the direct effects and and feedbacks that are illustrated - perhaps expanding this and cutting 1B, which just seems to be a picture with no real information content. (Maybe the annotation got lost in the proof generation but I don't see any at all on 1B.) Either way, given the central importance of good figures to communicating concepts in this kind of paper, I think fig 1 could do a lot more to really set out some specific feedbacks. (As an aside - separating figures from legends makes it very hard to interpret these kinds of schematic figures which rely heavily on legends. Maybe this is journal policy at submission, but as a courtesy to reviewers I'd recommend ignoring that and keeping figures and legends together, in the hope that this will eventually drive journal policies to change...)

L101 - Box 2 is referenced here, I don't have a Box 2 in my proof. Should this be Box 1?

Section 2 is very clear on what feedbacks are and why they're important. Fig 2 I like - it's clear and illustrates various feedbacks well - though I think the down arrow from Density to Growth Rate in B should have a + in it, not a -. Also unclear what the red question mark signifies in C. In the text, the specific examples I think need a little more work to really convince. The pollinator example has the following statements (L146): "The abundance of pollinators is known to increase the abundance of plants by facilitating plant reproduction. Higher pollinator diversity can

enhance plant diversity when there are positive interactions between different plant and pollinator species. Through this positive feedback, humans benefit when the plants are of cultural or agricultural value." Maybe this is all uncontentious but I would like to have seen some supporting evidence cited here (especially in a review paper). In particular, this is really a positive feedback (i.e. increasing pollinator abundance drives increases in plant diversity which in turn increases pollinator abundance, and so on - though presumably not without limit?) I think also the next sentence ("Human activities such as some agricultural practices and land use change have dramatically reduced pollinator abundance and diversity (46,47), causing humans to lose value in crop yields, and in turn motivating conservation and management actions.") would benefit from supporting evidence for the second two statements, and probably a bit more work to bring this back to the concept of feedbacks (negative here, right, as people do bad things which reduce pollinators, leading to poor crops, so influencing people to do better things to benefit pollinators?) Given that the next section states (L168) that "There is little mention [in IPBES] of full feedback cycles between biodiversity and ecosystem functioning", I think that really explicitly laying out such a full feedback cycle in section 2 would be useful.

Section 4 I thought was excellent - these really are key knowledge gaps, and it's really useful to have them set out so clearly here.

I was slightly less convinced by Section 5 - mainly, the distinction between key knowledge gaps and grand research challenges seems a little contrived to me, and I felt that Sections 4 and 5 could potentially be integrated. For instance, Challenge 2 states "we do not have a robust model defining how changes in biodiversity at large scales (e.g., global or continental) interact with changes at fine spatial scales (e.g., locally operating processes such as disturbance, invasion or restoration) to influence biodiversity and ecosystem functioning" - yes, this is a really good point. But why is that designated a grand challenge rather than a key knowledge gap? While Challenge 6 felt it would likely fit better into the Agenda for Action (section 6) as there are some quite clear recommendations that are (or could be) made to meet that challenge.

Some of the challenges are well articulated (Challenge 5 was particularly good I thought) but others don't really seem to contain much that's concrete - challenge 1, for example, "Meeting this challenge requires transdisciplinary scholarship to identify the most important feedbacks, as well as to develop approaches to model these feedbacks. The models and concepts must be tested and explored with theory and experiments. Including human systems in our understanding of the biosphere is not only a scientific but also philosophical challenge" - fine, I don't disagree; but I don't think there's anything that anyone could disagree with, it's rather empty and the kind of grand statement I feel like I've read dozens of times over the years. What is it really saying?

Section 6 I also found rather generic and lacking in the kind of specific detail which would really set it apart from similar calls for action. For instance, Dornelas et al. (2019 [https://doi-org.sheffield.idm.oclc.org/10.1111/geb.13025](https://doi.org/sheffield.idm.oclc.org/10.1111/geb.13025)) made some similar kinds of appeals in their paper on a 'macroscope' for coordinated collection of macroecological data, which is of course different (but clearly related) to the current agenda. They set out some ideas for achieving this as an 'appeal' rather than an 'agenda' - I feel like an agenda calls for more actionable points. Do the authors envisage this agenda being implemented under IPBES? Or something else? What kind of investment, and from whom, are they envisaging? I get that a purpose of this paper is to set out this agenda which can then be leveraged with funding agencies but I'm afraid this section just read a little high level and lacking in practicalities for it to really make for a useful agenda.

The point about relevant data not being curated in central databases (L410) is a good one, although more than GenBank exists - see e.g. Webb & Vanhoorne <https://doi.org/10.1098/rstb.2019.0445> for attempts to define linkages between existing data repositories relevant to biodiversity; and I think things like the Bari manifesto are key here too - lots of progress has been made by biodiversity informaticians (see Hardisty et al. <https://doi.org/10.1016/j.ecoinf.2018.11.003>) and it is important to build on, rather than reinvent, such initiatives.

Review form: Reviewer 2

Recommendation

Major revision is needed (please make suggestions in comments)

Scientific importance: Is the manuscript an original and important contribution to its field?

Good

General interest: Is the paper of sufficient general interest?

Good

Quality of the paper: Is the overall quality of the paper suitable?

Marginal

Is the length of the paper justified?

Yes

Should the paper be seen by a specialist statistical reviewer?

No

Do you have any concerns about statistical analyses in this paper? If so, please specify them explicitly in your report.

No

It is a condition of publication that authors make their supporting data, code and materials available - either as supplementary material or hosted in an external repository. Please rate, if applicable, the supporting data on the following criteria.

Is it accessible?

N/A

Is it clear?

N/A

Is it adequate?

N/A

Do you have any ethical concerns with this paper?

No

Comments to the Author

Review RSPB-2021-0783

I like the ambition of this paper very much. The authors have identified important issues of scientific, policy and societal concern in the ways that biodiversity-relevant sciences theorise, analyse, model, monitor and communicate about today's linked social and ecological changes.

But the paper does not engage clearly enough with these challenge areas. In brief, improvements are needed on the description of system concepts and methods and their real-world use and implications; the academic and policy debates around "ecosystem services" and their relation to people's understandings of nature and the processes of human-driven environmental change; and the rationale and structuring of the questions, challenges, action items especially regarding the conceptual framings implied in the figures.

Specific comments:

A. The paper relies heavily on system concepts and methods, but is often unclear about what the system of interest actually is, and despite introducing the B-E-H system idea the manuscript often also slips between systems with people in and some notional ideal “natural” system.

The strong heading II statement – “feedbacks drive relationships” is true enough as a generic description of dynamic systems, but it ends up as a strange representation of human activities as drivers of change and reconfigurers of living systems.

Line 103 flags some methodological issues where feedbacks feature in quite diverse ways in interaction networks, system dynamics / complex systems theory, and as a metaphor. It would be helpful for the paper to clearly articulate whether and in which contexts these feedbacks are observable / empirical behaviours of living nature itself (including people), ways of conceptualising and representing those behaviours (epistemologically), or ways of communicating complexity and dynamic risks.

Synergy gives a more than additive effect because things from outside of the system are shaping the behaviour.

Line 108 - Complex adaptive system outcomes cannot be predicted, by definition – but their tendencies, properties and behaviours can be characterised.

Line 120 – model feedbacks – there are many approaches other than the system dynamics modelling approach set out here (eg, agent based models, some kinds of network analyses as hinted earlier). In other fields, modelling parametrised dynamics allows for abductive analysis of feedbacks, also using multi model comparison, etc. p8 lines 169-174 – full feedback cycles in models are not mentioned for good reason. It is perfectly ok for feedbacks to be identified as an outcome of some kinds of model. Line 190 – “getting feedbacks right in our models” Is a very narrow description of what needs to be done. Understanding feedbacks does not always mean encoding them as feedbacks in models. The framing here makes it seem as if all that needs to be done is make fresh new models with feedbacks in, and then the system will be nicely deterministic and predictable. It would be good to expand more on the model use process – who believes model outputs? In what contexts are deterministic model outputs being used for complex / chaotic systems?

P9 Line 194 – this is a problematic question because the human activities are invisible in the framing. How do human caused destruction and simplification of other life and living assemblages influence future evolution? Future scope for further human caused destruction and simplification?? “Furthermore, we need to distinguish when positive vs negative feedbacks dominate if they require very different management actions.” – this text reverts to narrow system dynamics terminology, and what has become rather a trope (positive feedbacks bad, negative feedbacks good). Consider the implications of biodiversity as an open system, as a coupled system. Are managers going to understand a system-dynamics framed story of human devastation of living nature better than other framings? Is accurate prediction really the aim?

B. Put the scientist into the system – eg in Lines 407-412, glossary.

C. The paper contains several listicles: 5 gaps, 6 challenges, 5 agenda points... – How exactly and why are they defined? Who for?

GAPS:

Gap 2, p9 – future trajectories are impossible to predict with observations of current wellbeing and ecosystem functioning too. Be more concrete here if possible. Monitoring programs must be targeted at particular issues (we can’t meaningfully watch everything all of the time) – so there are challenges and opportunities for complementarity of programs, development of protocols (including for metadata) that can help bridge different epistemic communities, and of course synthesis. Be careful not to come across as an uncritical climate-technoscience fan – D&A by statistics might well work for deterministic systems, but there are well-placed methodological critiques to consider for living systems, and the question of how measurement itself narrows perspectives is particularly acute. This is at the heart of community-based monitoring debates – a lived environment can be understood profoundly differently than a remotely monitored one. Even multi-expert approaches (eg all-taxa inventories) can be useful in demonstrating the

importance of pluralism and diverse perspectives in understanding what may well be irreducible complexity.

Gap 3 BEF effects hinge critically on definitions (of both the terms themselves and the systems of interest), which are too often unclear. There is something back to front in the argument for this gap – observing change in terms of partial and “linear” studies is a large part of understanding the range of dynamics (back to all-taxa studies that may well prompt awe of the real complexity of life!). Can you expand on how and why refs 40 and 62 support this point? Can’t systemic synthesis provide ways to using the hundreds of experiments rather than bluntly stating they “cannot be used”?

Gap 4 – The IPBES statement needs to be picked over not just cited! It is full of SES jargon. The B-E-H system idea of the paper could be linked to this point, but it is currently an undefined blob interacting with a contested blob interacting with an undifferentiated blob. One of the battlegrounds of IPBES is to maintain diversity and pluralism in and alongside biodiversity scenarios. If “understanding human-biodiversity feedbacks” means bringing all knowledge into these blobs so that they can be modelled, then the insights about feedbacks may become more impoverished not less.

CHALLENGES:

Line 249 - science based? Is it scientific or not? If science is just the basis what else happens before it becomes understanding? Yes, it’s a buzz-phrase.

Line 251-4 why six challenges, where do they come from? How do they relate to SES, the BEH system, BEF, etc?

Challenge 1: “identify”? The way people choose to represent the system determines the feedbacks. “The goal is to fully integrate human components of feedbacks” – is this not an ambition for a model that is the same as the territory? Whose goal is such a system model /concept?

Line 261 Including human systems in our understanding of the biosphere is not just a scientific and philosophical but an ethical challenge.

Challenge 2: I can’t see the distinction between challenges 1 and 2. Here it is stated that the focus is coupled human-natural systems – but doesn’t the B-E-H system allow for happy humans? This text also appears to focus on a network(ed) methodology for complex adaptive systems, rather than system dynamics implicit in challenge 1 – but it also gets into metaphor (links between feedbacks). 271-275 is a problem statement that would be classic “mixed methods” in social sciences – is this the kind of challenge meant here?

Challenge 3: “different dimensions of biodiversity...” Is this in reference to the rather rigid and artificial categories in the CBD text? The listing of dimensions in line 280 are still all countable things, rather than an ecosystemic view of biodiversity or much less a bioculturally inclusive view.

*Estimate future states of the biosphere” – assumptions about predictability of future life hinge critically on whether the difference between complex and simple systems is a difference in kind or in degree. If they are categorically different in kind, then the basis for “estimating” future states (other than human-destroyed!) is really open. Species over time – should this not consider tokogeny not just phylogeny, which specific organisms live and reproduce.

Challenge 4: what people use and want depends on what they can get. “However, the strengths of interactions between biodiversity and services remains less established for many services, especially with respect to the role of biodiversity-ecosystem functioning feedbacks as defined more strictly to be additional to the contributions of particular species” “Furthermore, the dependence of services upon feedbacks between biodiversity and ecosystem functions is not well characterized.” For meaningful system analysis, components in a system must be. They can’t be multiple, or sometimes be, or be only when viewed by a particular viewer – so the points made in this challenge imply that systems methods and feedbacks analysis may well be entirely inappropriate for these kinds of issues.

Challenge 5 – is about sensing change and adaptive systems. The two lines “Furthermore, feedbacks are difficult to detect from most observational datasets because they require coordinated observations of several facets- of a system” and Line 323- “yet these perspectives cannot be robustly integrated into models of change over time without accompanying theory and

empirical evidence for relationships between observations and biological processes” presuppose some kind of stationarity in the system. The real-world challenge is rather how can feedbacks and systemic understanding help us (scientists, the world at large) to do better with the knowledge that we can NEVER observe all facets of a living, self-reorganizing complex system all the time? Challenge 6 - Line 330-333 Give examples here, to prevent just sounding woo. Even within specialist communities, people speak at cross purposes about complexity and feedbacks. And they use jargon without always having clear substance behind, eg, invoking boundaries that are more subjective than empirical, thresholds that move around, tipping points that are linear responses, etc. “Feedbacks can guide decisions about how to invest observation effort, about prioritization of conservation actions to vulnerable or stable systems, and in optimal workflows to convert knowledge into action to protect future biodiversity.” How can feedbacks do this?

ACTION ITEMS

Line 338 is vitally important, but appears rather late – contrasting with Gap 2 and Challenge 2. See also line 237-238 – traditional knowledge doesn’t count in the “persistent perception”?

Line 354-356 – this could provide a critical framing of the paper, part of the rationale for the B-E-H idea in the first place.

Line 366 → Action Item 2 - a key focus here is also to enable biosphere models to link to other kinds of global models. For example, LeClere et al 2020 shows how different kinds of biodiversity models can be used in combination with different kinds of climate impact models (IAMs more generally), giving better insights about multidimensional / multi-driver environmental change

Item 3 Observe biodiversity, ecosystem functioning and human activity change together: we are observing these changes together. The issue is as much about scientific disaggregation.

New statistical approaches, comprehensive, Kühl et al 2020: There are good intentions in this section that recognise the history of colonialisms and exploitation, but still take care to avoid hubris overall – there is a tone of “when everyone sees the world as we do...”

Item 4: It would be interesting to compare with atmospheric sciences: “observational programs should be guided by theory that includes feedbacks, and coupled with experimental programs to understand feedbacks.”

Item 5 – Identify a leadership team from where?! Under what aegis? There used to be Diversitas. Any lessons from that and similar international networks?

“A leadership team must assemble, must be able to draw on existing scientific knowledge and work with the research community to develop research programs” What would be the consequences instead of networked working, without head honchos “drawing on” other people’s work? What other kinds of social organization could be thought of?

Conclusion – it is over 30 years since biodiversity became a thing. But feedbacks of human wellbeing and life have a much longer history.

To meet the policy challenges of the coming decades – be specific about these earlier in the paper. Line 425 - “We risk making decisions based on modeled futures that do not capture the full range of likely possibilities (21,50,53,65,85,86).” This sentence has lots of refs tacked on at the end – argue a clear case from these studies / opinions earlier in the paper.

Other comments:

Glossary: Fix the ecosystem services / NCP tangle. If the contested nature of the concept matters, explain this properly in the paper itself. Ecosystem services as part of the function-service-benefit chain would be coherent with the B-E-H system presented in the paper. These diverse framings are perhaps a central part of the challenge that feedbacks modelling encounters.

Figure 1 – unclear diagrams, low quality figures. The article is not a picture book – its needs scientific visualizations. The Venn diagram doesn’t make sense. Ecosystem services in the “humans” system, biodiversity and ecosystem function in the “nature” system

Figure 2 is about closed and open systems, or systems with defined boundaries. It also shows a switch between hypothesis-led science and other framings of inquiry.

Figure 3 – Feedbacks also relate to system scale – degree of connection, nodality

Abstract P5 - formally, feedbacks are a feature not a component

P6 heading - Dynamic feedbacks are causes and consequences

Line 68 – defining biodiversity – from genes to social-ecological systems and beyond...

Line 70 – functioning, services, NCP – glossary points out these are different worldviews ontologies.

P7 Line 107 – feedbacks make human and natural systems change in non-intuitive ways?

Coupled systems are of interest here. The literature cited is about feedbacks in modelling to improve physical climate / biophysical Earth system representation. CHANS are still outside this system.

Big statement line 126 – explain more simply – feedbacks explain change and stability.

Line 136 – disease can have positive feedbacks – pandemics are obvious example. First define the system as a diverse system at the start of the paragraph.

P8 line 152 – explain clearer. The simple story is bad ag and land use practices cause lower crop yields and economic losses (value). Adding a biodiversity component to the story explains why just changing practices and replanting the land won't necessarily fix the problem – if pollinators have been driven away / to extinction. THEN it becomes clearer that the role of biodiversity in feedbacks has not been made clear to policy people.

Decision letter (RSPB-2021-0783.R0)

30-Jul-2021

Dear Dr O'Connor:

Your manuscript has now been peer reviewed and the reviewers' comments (not including confidential comments to the Editor) are included at the end of this email for your reference. As you will see, the reviewers like the topic and ambition of your review, but have raised some concerns that need addressing before it could be considered acceptable. So, I would like to invite you to revise your manuscript to address their concerns. The key points, to paraphrase reviewer 2, are: to 1) tackle the treatment of systems concepts and methods more explicitly; 2) clarify what the "policy challenges of the coming decades" are in a way that is consistent with the new systems feedbacks paradigm you want to promote; 3) provide clear definitions for their "system components" (biodiversity, ecosystem services, human wellbeing) - and more importantly, be upfront about the fact that each of these components is contested in science and policy alike; 4) explain how BEF and B-E-H provide structuring aspects for the paper; and 5) improve the figures so the content is clearer and more easily understood.

Comments to Author(s)

Research ethics:

Use of animals and field studies:

It is a condition of publication that you make available the data and research materials supporting the results in the article (<https://royalsociety.org/journals/authors/author-guidelines/#data>). Datasets should be deposited in an appropriate publicly available repository and details of the associated accession number, link or DOI to the datasets must be included in the Data Accessibility section of the article (<https://royalsociety.org/journals/ethics-policies/data-sharing-mining/>). Reference(s) to datasets should also be included in the reference list of the article with DOIs (where available).

Please submit a copy of your revised paper within three weeks. If we do not hear from you within this time your manuscript will be rejected. If you are unable to meet this deadline please let us know as soon as possible, as we may be able to grant a short extension.

Best wishes,
Innes Cuthill

Professor Innes Cuthill
Reviews Editor, Proceedings B,
mailto:proceedingsb@royalsociety.org

Reviewer(s)' Comments to Author:

Referee: 1

Comments to the Author(s)

I enjoyed reading this ms and I think it makes an important contribution, setting out clearly why feedbacks are important in B-E-H research, how they've been overlooked, and what could (and should) be done about this. Some parts are stronger than others - I think it builds really well to the end of Section 4, but then peters out a little and the final couple of sections are big on grand sweeping statements but lacking a little in specifics. However overall this is well conceived and contains some excellent points. Additional comments follow, more or less in the order in which they arise in the ms.

Section 1 convinces me of the importance of feedbacks and sets out the structure of the piece really clearly. However, I don't think it is very well supported by fig 1, which in my view could do a much better job of explicitly setting out the feedbacks. As it stands I found it rather difficult to follow, and to identify where feedbacks fit, what exactly they entail, and how they have typically been missed in B-E-H work. 1A could be better annotated I think to specifically identify the direct effects and feedbacks that are illustrated - perhaps expanding this and cutting 1B, which just seems to be a picture with no real information content. (Maybe the annotation got lost in the proof generation but I don't see any at all on 1B.) Either way, given the central importance of good figures to communicating concepts in this kind of paper, I think fig 1 could do a lot more to really set out some specific feedbacks. (As an aside - separating figures from legends makes it very hard to interpret these kinds of schematic figures which rely heavily on legends. Maybe this is journal policy at submission, but as a courtesy to reviewers I'd recommend ignoring that and keeping figures and legends together, in the hope that this will eventually drive journal policies to change...)

L101 - Box 2 is referenced here, I don't have a Box 2 in my proof. Should this be Box 1?

Section 2 is very clear on what feedbacks are and why they're important. Fig 2 I like - it's clear and illustrates various feedbacks well - though I think the down arrow from Density to Growth Rate in B should have a + in it, not a -. Also unclear what the red question mark signifies in C. In the text, the specific examples I think need a little more work to really convince. The pollinator example has the following statements (L146): "The abundance of pollinators is known to increase the abundance of plants by facilitating plant reproduction. Higher pollinator diversity can enhance plant diversity when there are positive interactions between different plant and pollinator species. Through this positive feedback, humans benefit when the plants are of cultural or agricultural value." Maybe this is all uncontentious but I would like to have seen some supporting evidence cited here (especially in a review paper). In particular, this is really a positive feedback (i.e. increasing pollinator abundance drives increases in plant diversity which in turn increases pollinator abundance, and so on - though presumably not without limit?) I think also the next sentence ("Human activities such as some agricultural practices and land use change have dramatically reduced pollinator abundance and diversity (46,47), causing humans to lose

value in crop yields, and in turn motivating conservation and management actions.”) would benefit from supporting evidence for the second two statements, and probably a bit more work to bring this back to the concept of feedbacks (negative here, right, as people do bad things which reduce pollinators, leading to poor crops, so influencing people to do better things to benefit pollinators?) Given that the next section states (L168) that “There is little mention [in IPBES] of full feedback cycles between biodiversity and ecosystem functioning”, I think that really explicitly laying out such a full feedback cycle in section 2 would be useful.

Section 4 I thought was excellent - these really are key knowledge gaps, and it's really useful to have them set out so clearly here.

I was slightly less convinced by Section 5 - mainly, the distinction between key knowledge gaps and grand research challenges seems a little contrived to me, and I felt that Sections 4 and 5 could potentially be integrated. For instance, Challenge 2 states “we do not have a robust model defining how changes in biodiversity at large scales (e.g., global or continental) interact with changes at fine spatial scales (e.g., locally operating processes such as disturbance, invasion or restoration) to influence biodiversity and ecosystem functioning” - yes, this is a really good point. But why is that designated a grand challenge rather than a key knowledge gap? While Challenge 6 felt it would likely fit better into the Agenda for Action (section 6) as there are some quite clear recommendations that are (or could be) made to meet that challenge.

Some of the challenges are well articulated (Challenge 5 was particularly good I thought) but others don't really seem to contain much that's concrete - challenge 1, for example, “Meeting this challenge requires transdisciplinary scholarship to identify the most important feedbacks, as well as to develop approaches to model these feedbacks. The models and concepts must be tested and explored with theory and experiments. Including human systems in our understanding of the biosphere is not only a scientific but also philosophical challenge” - fine, I don't disagree; but I don't think there's anything that anyone could disagree with, it's rather empty and the kind of grand statement I feel like I've read dozens of times over the years. What is it really saying?

Section 6 I also found rather generic and lacking in the kind of specific detail which would really set it apart from similar calls for action. For instance, Dornelas et al. (2019 [https://doi-org.sheffield.idm.oclc.org/10.1111/geb.13025](https://doi.org/sheffield.idm.oclc.org/10.1111/geb.13025)) made some similar kinds of appeals in their paper on a 'macroscope' for coordinated collection of macroecological data, which is of course different (but clearly related) to the current agenda. They set out some ideas for achieving this as an 'appeal' rather than an 'agenda' - I feel like an agenda calls for more actionable points. Do the authors envisage this agenda being implemented under IPBES? Or something else? What kind of investment, and from whom, are they envisaging? I get that a purpose of this paper is to set out this agenda which can then be leveraged with funding agencies but I'm afraid this section just read a little high level and lacking in practicalities for it to really make for a useful agenda.

The point about relevant data not being curated in central databases (L410) is a good one, although more than GenBank exists - see e.g. Webb & Vanhoorne <https://doi.org/10.1098/rstb.2019.0445> for attempts to define linkages between existing data repositories relevant to biodiversity; and I think things like the Bari manifesto are key here too - lots of progress has been made by biodiversity informaticians (see Hardisty et al. <https://doi.org/10.1016/j.ecoinf.2018.11.003>) and it is important to build on, rather than reinvent, such initiatives.

Referee: 2

Comments to the Author(s)

Review RSPB-2021-0783

I like the ambition of this paper very much. The authors have identified important issues of scientific, policy and societal concern in the ways that biodiversity-relevant sciences theorise, analyse, model, monitor and communicate about today's linked social and ecological changes.

But the paper does not engage clearly enough with these challenge areas. In brief, improvements are needed on the description of system concepts and methods and their real-world use and implications; the academic and policy debates around "ecosystem services" and their relation to people's understandings of nature and the processes of human-driven environmental change; and the rationale and structuring of the questions, challenges, action items especially regarding the conceptual framings implied in the figures.

Specific comments:

A. The paper relies heavily on system concepts and methods, but is often unclear about what the system of interest actually is, and despite introducing the B-E-H system idea the manuscript often also slips between systems with people in and some notional ideal "natural" system.

The strong heading II statement – "feedbacks drive relationships" is true enough as a generic description of dynamic systems, but it ends up as a strange representation of human activities as drivers of change and reconfigurers of living systems.

Line 103 flags some methodological issues where feedbacks feature in quite diverse ways in interaction networks, system dynamics / complex systems theory, and as a metaphor. It would be helpful for the paper to clearly articulate whether and in which contexts these feedbacks are observable / empirical behaviours of living nature itself (including people), ways of conceptualising and representing those behaviours (epistemologically), or ways of communicating complexity and dynamic risks.

Synergy gives a more than additive effect because things from outside of the system are shaping the behaviour.

Line 108 - Complex adaptive system outcomes cannot be predicted, by definition – but their tendencies, properties and behaviours can be characterised.

Line 120 – model feedbacks – there are many approaches other than the system dynamics modelling approach set out here (eg, agent based models, some kinds of network analyses as hinted earlier). In other fields, modelling parametrised dynamics allows for abductive analysis of feedbacks, also using multi model comparison, etc. p8 lines 169-174 – full feedback cycles in models are not mentioned for good reason. It is perfectly ok for feedbacks to be identified as an outcome of some kinds of model. Line 190 – "getting feedbacks right in our models" Is a very narrow description of what needs to be done. Understanding feedbacks does not always mean encoding them as feedbacks in models. The framing here makes it seem as if all that needs to be done is make fresh new models with feedbacks in, and then the system will be nicely deterministic and predictable. It would be good to expand more on the model use process – who believes model outputs? In what contexts are deterministic model outputs being used for complex / chaotic systems?

P9 Line 194 – this is a problematic question because the human activities are invisible in the framing. How do human caused destruction and simplification of other life and living assemblages influence future evolution? Future scope for further human caused destruction and simplification?? "Furthermore, we need to distinguish when positive vs negative feedbacks dominate if they require very different management actions." – this text reverts to narrow system dynamics terminology, and what has become rather a trope (positive feedbacks bad, negative feedbacks good). Consider the implications of biodiversity as an open system, as a coupled system. Are managers going to understand a system-dynamics framed story of human devastation of living nature better than other framings? Is accurate prediction really the aim?

B. Put the scientist into the system – eg in Lines 407-412, glossary.

C. The paper contains several listicles: 5 gaps, 6 challenges, 5 agenda points... – How exactly and why are they defined? Who for?

GAPS:

Gap 2, p9 – future trajectories are impossible to predict with observations of current wellbeing and ecosystem functioning too. Be more concrete here if possible. Monitoring programs must be targeted at particular issues (we can't meaningfully watch everything all of the time) – so there are challenges and opportunities for complementarity of programs, development of protocols

(including for metadata) that can help bridge different epistemic communities, and of course synthesis. Be careful not to come across as an uncritical climate-technoscience fan – D&A by statistics might well work for deterministic systems, but there are well-placed methodological critiques to consider for living systems, and the question of how measurement itself narrows perspectives is particularly acute. This is at the heart of community-based monitoring debates – a lived environment can be understood profoundly differently than a remotely monitored one. Even multi-expert approaches (eg all-taxa inventories) can be useful in demonstrating the importance of pluralism and diverse perspectives in understanding what may well be irreducible complexity.

Gap 3 BEF effects hinge critically on definitions (of both the terms themselves and the systems of interest), which are too often unclear. There is something back to front in the argument for this gap – observing change in terms of partial and “linear” studies is a large part of understanding the range of dynamics (back to all-taxa studies that may well prompt awe of the real complexity of life!). Can you expand on how and why refs 40 and 62 support this point? Can’t systemic synthesis provide ways to using the hundreds of experiments rather than bluntly stating they “cannot be used”?

Gap 4 – The IPBES statement needs to be picked over not just cited! It is full of SES jargon. The B-E-H system idea of the paper could be linked to this point, but it is currently an undefined blob interacting with a contested blob interacting with an undifferentiated blob. One of the battlegrounds of IPBES is to maintain diversity and pluralism in and alongside biodiversity scenarios. If “understanding human-biodiversity feedbacks” means bringing all knowledge into these blobs so that they can be modelled, then the insights about feedbacks may become more impoverished not less.

CHALLENGES:

Line 249 - science based? Is it scientific or not? If science is just the basis what else happens before it becomes understanding? Yes, it’s a buzz-phrase.

Line 251-4 why six challenges, where do they come from? How do they relate to SES, the BEH system, BEF, etc?

Challenge 1: “identify”? The way people choose to represent the system determines the feedbacks. “The goal is to fully integrate human components of feedbacks” – is this not an ambition for a model that is the same as the territory? Whose goal is such a system model / concept?

Line 261 Including human systems in our understanding of the biosphere is not just a scientific and philosophical but an ethical challenge.

Challenge 2: I can’t see the distinction between challenges 1 and 2. Here it is stated that the focus is coupled human-natural systems – but doesn’t the B-E-H system allow for happy humans? This text also appears to focus on a network(ed) methodology for complex adaptive systems, rather than system dynamics implicit in challenge 1 – but it also gets into metaphor (links between feedbacks). 271-275 is a problem statement that would be classic “mixed methods” in social sciences – is this the kind of challenge meant here?

Challenge 3: “different dimensions of biodiversity...” Is this in reference to the rather rigid and artificial categories in the CBD text? The listing of dimensions in line 280 are still all countable things, rather than an ecosystemic view of biodiversity or much less a bioculturally inclusive view.

*Estimate future states of the biosphere” – assumptions about predictability of future life hinge critically on whether the difference between complex and simple systems is a difference in kind or in degree. If they are categorically different in kind, then the basis for “estimating” future states (other than human-destroyed!) is really open. Species over time – should this not consider tokogeny not just phylogeny, which specific organisms live and reproduce.

Challenge 4: what people use and want depends on what they can get. “However, the strengths of interactions between biodiversity and services remains less established for many services, especially with respect to the role of biodiversity-ecosystem functioning feedbacks as defined more strictly to be additional to the contributions of particular species” “Furthermore, the dependence of services upon feedbacks between biodiversity and ecosystem functions is not well characterized.” For meaningful system analysis, components in a system must be. They can’t be

multiple, or sometimes be, or be only when viewed by a particular viewer – so the points made in this challenge imply that systems methods and feedbacks analysis may well be entirely inappropriate for these kinds of issues.

Challenge 5 – is about sensing change and adaptive systems. The two lines “Furthermore, feedbacks are difficult to detect from most observational datasets because they require coordinated observations of several facets of a system” and Line 323- “yet these perspectives cannot be robustly integrated into models of change over time without accompanying theory and empirical evidence for relationships between observations and biological processes” presuppose some kind of stationarity in the system. The real-world challenge is rather how can feedbacks and systemic understanding help us (scientists, the world at large) to do better with the knowledge that we can NEVER observe all facets of a living, self-reorganizing complex system all the time?

Challenge 6 - Line 330-333 Give examples here, to prevent just sounding woo. Even within specialist communities, people speak at cross purposes about complexity and feedbacks. And they use jargon without always having clear substance behind, eg, invoking boundaries that are more subjective than empirical, thresholds that move around, tipping points that are linear responses, etc. “Feedbacks can guide decisions about how to invest observation effort, about prioritization of conservation actions to vulnerable or stable systems, and in optimal workflows to convert knowledge into action to protect future biodiversity.” How can feedbacks do this?

ACTION ITEMS

Line 338 is vitally important, but appears rather late – contrasting with Gap 2 and Challenge 2.

See also line 237-238 – traditional knowledge doesn’t count in the “persistent perception”?

Line 354-356 – this could provide a critical framing of the paper, part of the rationale for the B-E-H idea in the first place.

Line 366 – Action Item 2 - a key focus here is also to enable biosphere models to link to other kinds of global models. For example, LeClere et al 2020 shows how different kinds of biodiversity models can be used in combination with different kinds of climate impact models (IAMs more generally), giving better insights about multidimensional / multi-driver environmental change

Item 3 Observe biodiversity, ecosystem functioning and human activity change together: we are observing these changes together. The issue is as much about scientific disaggregation.

New statistical approaches, comprehensive, Kühl et al 2020: There are good intentions in this section that recognise the history of colonialisms and exploitation, but still take care to avoid hubris overall – there is a tone of “when everyone sees the world as we do...”

Item 4: It would be interesting to compare with atmospheric sciences: “observational programs should be guided by theory that includes feedbacks, and coupled with experimental programs to understand feedbacks.”

Item 5 – Identify a leadership team from where?! Under what aegis? There used to be Diversitas. Any lessons from that and similar international networks?

“A leadership team must assemble, must be able to draw on existing scientific knowledge and work with the research community to develop research programs” What would be the consequences instead of networked working, without head honchos “drawing on” other people’s work? What other kinds of social organization could be thought of?

Conclusion – it is over 30 years since biodiversity became a thing. But feedbacks of human wellbeing and life have a much longer history.

To meet the policy challenges of the coming decades – be specific about these earlier in the paper. Line 425 - “We risk making decisions based on modeled futures that do not capture the full range of likely possibilities (21,50,53,65,85,86).” This sentence has lots of refs tacked on at the end – argue a clear case from these studies / opinions earlier in the paper.

Other comments:

Glossary: Fix the ecosystem services / NCP tangle. If the contested nature of the concept matters, explain this properly in the paper itself. Ecosystem services as part of the function-service-benefit chain would be coherent with the B-E-H system presented in the paper. These diverse framings are perhaps a central part of the challenge that feedbacks modelling encounters.

Figure 1 – unclear diagrams, low quality figures. The article is not a picture book – its needs scientific visualizations. The Venn diagram doesn't makes sense. Ecosystem services in the "humans" system, biodiversity and ecosystem function in the "nature" system

Figure 2 is about closed and open systems, or systems with defined boundaries. It also shows a switch between hypothesis-led science and other framings of inquiry.

Figure 3 – Feedbacks also relate to system scale – degree of connection, nodality

Abstract P5 - formally, feedbacks are a feature not a component

P6 heading - Dynamic feedbacks are causes and consequences

Line 68 – defining biodiversity – from genes to social-ecological systems and beyond...

Line 70 – functioning, services, NCP – glossary points out these are different worldviews ontologies.

P7 Line 107 – feedbacks make human and natural systems change in non-intuitive ways?

Coupled systems are of interest here. The literature cited is about feedbacks in modelling to improve physical climate / biophysical Earth system representation. CHANS are still outside this system.

Big statement line 126 – explain more simply – feedbacks explain change and stability.

Line 136 – disease can have positive feedbacks – pandemics are obvious example. First define the system as a diverse system at the start of the paragraph.

P8 line 152 – explain clearer. The simple story is bad ag and land use practices cause lower crop yields and economic losses (value). Adding a biodiversity component to the story explains why just changing practices and replanting the land won't necessarily fix the problem – if pollinators have been driven away / to extinction. THEN it becomes clearer that the role of biodiversity in feedbacks has not been made clear to policy people.

Author's Response to Decision Letter for (RSPB-2021-0783.R0)

See Appendix A.

Decision letter (RSPB-2021-0783.R1)

20-Sep-2021

Dear Dr O'Connor

I am pleased to inform you that your manuscript entitled "Grand challenges in biodiversity-ecosystem functioning research in the era of science-policy platforms require explicit consideration of feedbacks" has been accepted for publication in Proceedings B.

If you are likely to be away from e-mail contact during this period, let us know. Due to rapid publication and an extremely tight schedule, if comments are not received, we may publish the paper as it stands.

Data Accessibility section

Open access

You are invited to opt for open access via our author pays publishing model. Payment of open access fees will enable your article to be made freely available via the Royal Society website as soon as it is ready for publication. For more information about open access publishing please visit our website at http://royalsocietypublishing.org/site/authors/open_access.xhtml.

The open access fee is £1,700 per article (plus VAT for authors within the EU). If you wish to opt for open access then please let us know as soon as possible.

Paper charges

Sincerely,

Professor Innes Cuthill

Appendix A

THE UNIVERSITY OF BRITISH COLUMBIA

Department of Zoology
Faculty of Science
4200 - 6270 University Boulevard
Vancouver, BC V6T 1Z4

Phone 604 822 2131
Fax 604 822 2416
www.zoology.ubc.ca

Editor:

Your manuscript has now been peer reviewed and the reviewers' comments (not including confidential comments to the Editor) are included at the end of this email for your reference. As you will see, the reviewers like the topic and ambition of your review, but have raised some concerns that need addressing before it could be considered acceptable.

Thank you!

So, I would like to invite you to revise your manuscript to address their concerns. The key points, to paraphrase reviewer 2, are: to:

- 1) tackle the treatment of systems concepts and methods more explicitly;
- 2) clarify what the "policy challenges of the coming decades" are in a way that is consistent with the new systems feedbacks paradigm you want to promote;
- 3) provide clear definitions for their "system components" (biodiversity, ecosystem services, human wellbeing) - and more importantly, be up-front about the fact that each of these components is contested in science and policy alike;
- 4) explain how BEF and B-E-H provide structuring aspects for the paper; and
- 5) improve the figures so the content is clearer and more easily understood.

We have made the revisions as requested, and all have strengthened and clarified the paper. Following the specific suggestions of Reviewer 2 and the editor, we have streamlined and brought out a bit more the systems concepts as they support the arguments we make and how BEF and B-E-H are core structuring aspects of the paper (wording changes throughout, detailed below). We have clarified the policy challenges to which we refer [lines 73-74, 188, 194-199]*, and we have streamlined language around system components (e.g., feedbacks as features of systems, biodiversity as a system component, etc) as well as their science and policy contexts. We have revised figure 1 to make more clear the direct, indirect effects and feedbacks.

*line numbers refer to the PDF with tracked changes; the line numbers differ in the Word documents and the pdf, and they are not continuous in the Word document with track changes. This discontinuity appears to be a bug in MS Word, but we felt these line numbers would be most useful for this stage of review.

Reviewer 1:

I enjoyed reading this ms and I think it makes an important contribution, setting out clearly why feedbacks are important in B-E-H research, how they've been overlooked, and what could (and should) be done about this. Some parts are stronger than others - I think it builds really well to the end of Section 4, but then peters out a little and the final couple of sections are big on grand sweeping statements but lacking a little in specifics. However overall this is well conceived and contains some excellent points. Additional comments follow, more or less in the order in which they arise in the ms.

Section 1 convinces me of the importance of feedbacks and sets out the structure of the piece really clearly. However, I don't think it is very well supported by fig 1, which in my view could do a much better job of explicitly setting out the feedbacks. As it stands I found it rather difficult to follow, and to identify where feedbacks fit, what exactly they entail, and how they have typically been missed in B-E-H work. 1A could be better annotated I think to specifically identify the direct effects and feedbacks that are illustrated - perhaps expanding this and cutting 1B, which just seems to be a picture with no real information content. (Maybe the annotation got lost in the proof generation but I don't see any at all on 1B.) Either way, given the central importance of good figures to communicating concepts in this kind of paper, I think fig 1 could do a lot more to really set out some specific feedbacks. (As an aside - separating figures from legends makes it very hard to interpret these kinds of schematic figures which rely heavily on legends. Maybe this is journal policy at submission, but as a courtesy to reviewers I'd recommend ignoring that and keeping figures and legends together, in the hope that this will eventually drive journal policies to change...)

Response: We have revised figure 1 to break out the components of the system - the diversity, ecosystem functioning, and the human components - while highlighting the direct and indirect effects and feedbacks in more detail. We have kept the aquatic and pollination examples (now panels D and E) and added annotation for panel E. We have also placed the figures right after the legends.

L101 - Box 2 is referenced here, I don't have a Box 2 in my proof. Should this be Box 1?
Response: That was a typo, there is no box 2 (there used to be one). I have corrected it and it now refers to Figure 2.

Section 2 is very clear on what feedbacks are and why they're important. Fig 2 I like - it's clear and illustrates various feedbacks well - though I think the down arrow from Density to Growth Rate in B should have a + in it, not a -. Also unclear what the red question mark signifies in C. In the text, the specific examples I think need a little more work to really convince. The pollinator example has the following statements (L146): "The abundance of pollinators is known to increase the abundance of plants by facilitating plant reproduction. Higher pollinator diversity can enhance plant diversity when there are

positive interactions between different plant and pollinator species. Through this positive feedback, humans benefit when the plants are of cultural or agricultural value.” Maybe this is all uncontroversial but I would like to have seen some supporting evidence cited here (especially in a review paper). In particular, this is really a positive feedback (i.e. increasing pollinator abundance drives increases in plant diversity which in turn increases pollinator abundance, and so on - though presumably not without limit?) I think also the next sentence (“Human activities such as some agricultural practices and land use change have dramatically reduced pollinator abundance and diversity (46,47), causing humans to lose value in crop yields, and in turn motivating conservation and management actions.”) would benefit from supporting evidence for the second two statements, and probably a bit more work to bring this back to the concept of feedbacks (negative here, right, as people do bad things which reduce pollinators, leading to poor crops, so influencing people to do better things to benefit pollinators?) Given that the next section states (L168) that “There is little mention [in IPBES] of full feedback cycles between biodiversity and ecosystem functioning”, I think that really explicitly laying out such a full feedback cycle in section 2 would be useful.

Response: Thank you, we have fixed the sign in Figure 2, and changed the legend to explain the red question mark. We have also expanded on the pollinator example, adding references and a few sentences [lines 164-184], and illustration in Figure 1E.

Section 4 I thought was excellent - these really are key knowledge gaps, and it’s really useful to have them set out so clearly here.

Super, thank you!

I was slightly less convinced by Section 5 - mainly, the distinction between key knowledge gaps and grand research challenges seems a little contrived to me, and I felt that Sections 4 and 5 could potentially be integrated. For instance, Challenge 2 states “we do not have a robust model defining how changes in biodiversity at large scales (e.g., global or continental) interact with changes at fine spatial scales (e.g., locally operating processes such as disturbance, invasion or restoration) to influence biodiversity and ecosystem functioning” - yes, this is a really good point. But why is that designated a grand challenge rather than a key knowledge gap? While Challenge 6 felt it would likely fit better into the Agenda for Action (section 6) as there are some quite clear recommendations that are (or could be) made to meet that challenge.

Response: Thank you for this suggestion. We did integrate these two sections now into a single section of knowledge gaps. We were able to consolidate three, so we now have a total of 7 knowledge gaps. Previous Challenge #6, about relating science to policy, indeed fits better as a prelude to the Agenda section, so was moved there.

Some of the challenges are well articulated (Challenge 5 was particularly good I thought) but others don't really seem to contain much that's concrete - challenge 1, for example, "Meeting this challenge requires transdisciplinary scholarship to identify the most important feedbacks, as well as to develop approaches to model these feedbacks. The models and concepts must be tested and explored with theory and experiments. Including human systems in our understanding of the biosphere is not only a scientific but also philosophical challenge" - fine, I don't disagree; but I don't think there's anything that anyone could disagree with, it's rather empty and the kind of grand statement I feel like I've read dozens of times over the years. What is it really saying?

Response: Challenge 1 has now been integrated with another to make the single point that we have a gap in our knowledge about what the feedbacks are (now Gap #5).

Section 6 I also found rather generic and lacking in the kind of specific detail which would really set it apart from similar calls for action. For instance, Dornelas et al. (2019 <https://doi-org.sheffield.idm.oclc.org/10.1111/geb.13025>) made some similar kinds of appeals in their paper on a 'macroscope' for coordinated collection of macroecological data, which is of course different (but clearly related) to the current agenda. They set out some ideas for achieving this as an 'appeal' rather than an 'agenda' - I feel like an agenda calls for more actionable points. Do the authors envisage this agenda being implemented under IPBES? Or something else? What kind of investment, and from whom, are they envisaging? I get that a purpose of this paper is to set out this agenda which can then be leveraged with funding agencies but I'm afraid this section just read a little high level and lacking in practicalities for it to really make for a useful agenda.

Response: Thank you, this was a helpful perspective to read. We have strengthened the language in the 'action items' to make the calls more clear and actionable.

The point about relevant data not being curated in central databases (L410) is a good one, although more than GenBank exists - see e.g. Webb & Vanhoorne <https://doi.org/10.1098/rstb.2019.0445> for attempts to define linkages between existing data repositories relevant to biodiversity; and I think things like the Bari manifesto are key here too - lots of progress has been made by biodiversity informaticians (see Hardisty et al. <https://doi.org/10.1016/j.ecoinf.2018.11.003>) and it is important to build on, rather than reinvent, such initiatives.

Response: Thank you for the suggestion of these references, these are perfect and we have included them.

Referee: 2

Comments to the Author(s)

Review RSPB-2021-0783

I like the ambition of this paper very much. The authors have identified important issues of scientific, policy and societal concern in the ways that biodiversity-relevant sciences theorise, analyse, model, monitor and communicate about today's linked social and ecological changes.

But the paper does not engage clearly enough with these challenge areas. In brief, improvements are needed on the description of system concepts and methods and their real-world use and implications; the academic and policy debates around "ecosystem services" and their relation to people's understandings of nature and the processes of human-driven environmental change; and the rationale and structuring of the questions, challenges, action items especially regarding the conceptual framings implied in the figures.

Response: Thank you for the positive feedback and the suggestions. We have followed these general suggestions in our revision by revising figure 1 (as also suggested by Reviewer 1), addressing more directly (though briefly) the debates around ecosystem services, and we have consolidated the knowledge gaps and challenges into a single section. We have also clarified the system concepts and methods, though for this too we had limited space with which to work while remaining within the word limits of the journal.

Specific comments:

A. The paper relies heavily on system concepts and methods, but is often unclear about what the system of interest actually is, and despite introducing the B-E-H system idea the manuscript often also slips between systems with people in and some notional ideal "natural" system.

Response: We have revised with this in mind. For example:

- **We revised the first paragraph in lines 63-74 to include more of the interdependence of biodiversity and human activities,**
- **Paragraph 2, line 79, we now refer to the 'B-E-H system components'**
- **Second line of paragraph 3 (line 91) explicitly references the three system components (diversity, ecosystems and humans).**

The strong heading II statement – "feedbacks drive relationships" is true enough as a generic description of dynamic systems, but it ends up as a strange representation of human activities as drivers of change and reconfigurers of living systems.

Response: This heading [line 109] has been revised as suggested below, and the point about the role of humans has also been addressed as noted above (line 114). Specifically, in this section, we revised the first sentence to clarify that the three components are part of one system (rather than two systems), and we removed a distinction between human and natural systems later in that paragraph (line 120).

Line 103 flags some methodological issues where feedbacks feature in quite diverse ways in interaction networks, system dynamics /complex systems theory, and as a metaphor. It would be helpful for the paper to clearly articulate whether and in which contexts these feedbacks are observable / empirical behaviours of living nature itself (including people), ways of conceptualising and representing those behaviours (epistemologically), or ways of communicating complexity and dynamic risks.

Response: This is an excellent point. We have added this distinction explicitly in the paragraph (around 134-136, 158-161) to highlight the different ways of using the concept. We were not able to fully address this point throughout the manuscript, given the space constraints we are facing.

Synergy gives a more than additive effect because things from outside of the system are shaping the behaviour.

Response: Yes, this makes sense. We did not see a place to revise the paper in respond to this point.

Line 108 - Complex adaptive system outcomes cannot be predicted, by definition – but their tendencies, properties and behaviours can be characterised.

Response: We agree; we acknowledge this by changing the word ‘predict’ in this context to ‘project’ at line 246

Line 120 – model feedbacks – there are many approaches other than the system dynamics modelling approach set out here (eg, agent based models, some kinds of network analyses as hinted earlier). In other fields, modelling parametrised dynamics allows for abductive analysis of feedbacks, also using multi model comparison, etc. p8 lines 169-174 – full feedback cycles in models are not mentioned for good reason. It is perfectly ok for feedbacks to be identified as an outcome of some kinds of model. Line 190 – “getting feedbacks right in our models” Is a very narrow description of what needs to be done. Understanding feedbacks does not always mean encoding them as feedbacks in models. The framing here makes it seem as if all that needs to be done is make fresh new models with feedbacks in, and then the system will be nicely deterministic and predictable. It would be good to expand more on the model use process – who believes model outputs?

In what contexts are deterministic model outputs being used for complex / chaotic systems?

Response: This is a good point, and to address it within the length constraints of the paper, we revised the text in these areas to be a bit more general (lines 135, 224-226) and avoid implying a specific approach to modeling feedbacks.

P9 Line 194 – this is a problematic question because the human activities are invisible in the framing. How do human caused destruction and simplification of other life and living assemblages influence future evolution? Future scope for further human caused destruction and simplification?? “Furthermore, we need to distinguish when positive vs negative feedbacks dominate if they require very different management actions.” – this text reverts to narrow system dynamics terminology, and what has become rather a trope (positive feedbacks bad, negative feedbacks good). Consider the implications of biodiversity as an open system, as a coupled system. Are managers going to understand a system-dynamics framed story of human devastation of living nature better than other framings? Is accurate prediction really the aim?

Response: These are excellent questions. To accommodate this point, we have revisited this ‘knowledge gap’ – now #1, line 243 – to discuss biodiversity change in the context of a component of a human-ecosystem-biodiversity system. We removed the word ‘predict’ and we removed the point about positive vs negative feedbacks.

B. Put the scientist into the system – eg in Lines 407-412, glossary.

Response: We did not understand how to revise following this comment. The text on the indicated lines (now lines 539-546) identifies current obstacles (open science, technological integration) and we left that as it was. We revised the glossary with regard to ecosystem services, NCP etc, so we added one line here to identify that NCP approaches integrating people into the system.

C. The paper contains several listicles: 5 gaps, 6 challenges, 5 agenda points... – How exactly and why are they defined? Who for?

Response: We have consolidated the gaps and challenges into one list, and further distinguished these two lists as 1) gaps to be tackled through research and 2) how exactly to do that as a community.

GAPS:

Gap 2, p9 – future trajectories are impossible to predict with observations of current wellbeing and ecosystem functioning too. Be more concrete here if possible. Monitoring programs must be targeted at particular issues (we can’t meaningfully watch everything all of the time) – so there are challenges and opportunities for complementarity of programs, development of protocols (including for metadata) that can help bridge

different epistemic communities, and of course synthesis. Be careful not to come across as an uncritical climate-technoscience fan – D&A by statistics might well work for deterministic systems, but there are well-placed methodological critiques to consider for living systems, and the question of how measurement itself narrows perspectives is particularly acute. This is at the heart of community-based monitoring debates – a lived environment can be understood profoundly differently than a remotely monitored one. Even multi-expert approaches (eg all-taxa inventories) can be useful in demonstrating the importance of pluralism and diverse perspectives in understanding what may well be irreducible complexity.

Response: Another excellent point. We have tried (and further clarified our language) to focus on projections and scenarios rather than predictions, to avoid the suggestion that we are seeking specific and precise predictions. We agree with the other points here about monitoring and purpose, but have chosen to keep this gap focused on the need for joint monitoring of diversity, function and human activities.

Gap 3 BEF effects hinge critically on definitions (of both the terms themselves and the systems of interest), which are too often unclear. There is something back to front in the argument for this gap – observing change in terms of partial and “linear” studies is a large part of understanding the range of dynamics (back to all-taxa studies that may well prompt awe of the real complexity of life!). Can you expand on how and why refs 40 and 62 support this point? Can’t systemic synthesis provide ways to using the hundreds of experiments rather than bluntly stating they “cannot be used”?

Response: True that isolating the direct effects is important; but identifying these alone does not reveal how the dynamics proceed over time in a system that allows for temporal feedbacks. We expanded the last sentence to explain this, and now have expanded figure 1 to support this argument. We also to refer to Figure 2 and the importance of including self-dependent feedbacks. Further, we removed these two references from this sentence; they illustrate the types of feedbacks that are not included, but they do not exactly make the point we are making.

Gap 4 – The IPBES statement needs to be picked over not just cited! It is full of SES jargon. The B-E-H system idea of the paper could be linked to this point, but it is currently an undefined blob interacting with a contested blob interacting with an undifferentiated blob. One of the battlegrounds of IPBES is to maintain diversity and pluralism in and alongside biodiversity scenarios. If “understanding human-biodiversity feedbacks” means bringing all knowledge into these blobs so that they can be modelled, then the insights about feedbacks may become more impoverished not less.

Response: We specifically linked B-E-H to SES, as suggested, and tightened the language around how we are suggesting we need to integrate understanding of human - biodiversity relationships: “1) The challenge we face is therefore to integrate the multiple human (behavioral, demographic, social, political, economic,

institutional) components of the B-E-H system in ways that reflect the dependence of human wellbeing on biodiversity as well as the effects of humans on biodiversity (26,65). Meeting this challenge requires transdisciplinary scholarship to identify the dominant feedbacks and the feedbacks of particular interest to stakeholders, as well as to develop approaches to model these feedbacks and to communicate their effects on system projections and scenarios.”

CHALLENGES:

Line 249 - science based? Is it scientific or not? If science is just the basis what else happens before it becomes understanding? Yes, it's a buzz-phrase.

Response: We removed this term from this part of the paper. We kept it in reference to IPBES (Section III), it is our understanding that this is the preferred term for the policy platform, because the platform includes science but is not always scientific when it includes other knowledge frameworks such as indigenous knowledge.

Line 251-4 why six challenges, where do they come from? How do they relate to SES, the BEH system, BEF, etc?

Response: These emerged from our review in the preceding parts of the paper. We have now merged them with the knowledge gaps, for a total of 7. We have added this sentence to the first paragraph of this section [line 228]: “Our survey revealed seven knowledge gaps that emerge when we considered the B-E-H system as a whole system, rather than take previously prevalent perspectives that emphasize two of the three components – Biodiversity and Ecosystem Function (BEF) that tends to consider human activities as outside the system, or socioecological systems (SES) in which biodiversity and functioning are lumped into one component.”

Challenge 1: “identify”? The way people choose to represent the system determines the feedbacks. “The goal is to fully integrate human components of feedbacks” – is this not an ambition for a model that is the same as the territory? Whose goal is such a system model /concept?

Response: this challenge has now been merged with a knowledge gap, and is now Gap #5. We removed the word identify, as well as ‘fully’, and sharpened this to specifically target human activities.

Line 261 Including human systems in our understanding of the biosphere is not just a scientific and philosophical but an ethical challenge.

Response: We ended up removing this sentence, in an effort to keep the word count down.

Challenge 2: I can't see the distinction between challenges 1 and 2. Here it is stated that the focus is coupled human-natural systems – but doesn't the B-E-H system allow for happy humans? This text also appears to focus on a network(ed) methodology for complex adaptive systems, rather than system dynamics implicit in challenge 1 – but it also gets into metaphor (links between feedbacks). 271-275 is a problem statement that would be classic “mixed methods” in social sciences – is this the kind of challenge meant here?

Response: The distinction is that challenge 2 (now gap #3) is specifically about how we can consider feedbacks when we are dealing with observations from different scales and resolutions. We have added a sentence to clarify that this gap is particularly about scale dependence [line 284].

Challenge 3: “different dimensions of biodiversity...” Is this in reference to the rather rigid and artificial categories in the CBD text? The listing of dimensions in line 280 are still all countable things, rather than an ecosystemic view of biodiversity or much less a bioculturally inclusive view.

Response: This is now knowledge gap # 6. This is specifically in reference to scientific measures of diversity, and not the latter, more pluralistic perspective. We have clarified this with more specific terms.

*Estimate future states of the biosphere” – assumptions about predictability of future life hinge critically on whether the difference between complex and simple systems is a difference in kind or in degree. If they are categorically different in kind, then the basis for “estimating” future states (other than human-destroyed!) is really open. Species over time – should this not consider tokogeny not just phylogeny, which specific organisms live and reproduce.

Response: An excellent point; we actually removed this sentence as we try to keep within the word limit.

Challenge 4: what people use and want depends on what they can get. “However, the strengths of interactions between biodiversity and services remains less established for many services, especially with respect to the role of biodiversity-ecosystem functioning feedbacks as defined more strictly to be additional to the contributions of particular species” ”Furthermore, the dependence of services upon feedbacks between biodiversity and ecosystem functions is not well characterized.” For meaningful system analysis, components in a system must be. They can't be multiple, or sometimes be, or be only when viewed by a particular viewer – so the points made in this challenge imply that systems methods and feedbacks analysis may well be entirely inappropriate for these kinds of issues.

Response: We have removed this challenge altogether

Challenge 5 – is about sensing change and adaptive systems. The two lines “Furthermore, feedbacks are difficult to detect from most observational datasets because they require coordinated observations of several facets of a system” and Line 323- “yet these perspectives cannot be robustly integrated into models of change over time without accompanying theory and empirical evidence for relationships between observations and biological processes” presuppose some kind of stationarity in the system. The real-world challenge is rather how can feedbacks and systemic understanding help us (scientists, the world at large) to do better with the knowledge that we can NEVER observe all facets of a living, self-reorganizing complex system all the time?

Response: Yes, we completely agree. But this point would require a bit of space to make effectively, and we simply do not have that space here.

Challenge 6 - Line 330-333 Give examples here, to prevent just sounding woo. Even within specialist communities, people speak at cross purposes about complexity and feedbacks. And they use jargon without always having clear substance behind, eg, invoking boundaries that are more subjective than empirical, thresholds that move around, tipping points that are linear responses, etc. “Feedbacks can guide decisions about how to invest observation effort, about prioritization of conservation actions to vulnerable or stable systems, and in optimal workflows to convert knowledge into action to protect future biodiversity.” How can feedbacks do this?

Response: We didn’t feel we have the space to expand with examples, but we agree this is an important point to be clear on. We moved this out of the ‘gaps’ list, and instead used it as context for the Agenda for Action.

ACTION ITEMS

Line 338 is vitally important, but appears rather late – contrasting with Gap 2 and Challenge 2. See also line 237-238

– traditional knowledge doesn’t count in the “persistent perception”?

Response: Yes, we agree. At the mention of the ‘persistent perception’, we have clarified that this perception persists in western science framing of the B-E-H system. We have chosen to leave this point till late in the paper (now line 456-475), because the first half of the paper - sections I - IV - are restricted to the scientific framing of biodiversity, ecosystem functioning and human well-being. This is a necessary restriction to allow us to speak meaningfully to the issue in some depth, though clearly the trade-off is not being able to explore many ideas like the traditional knowledge links in sufficient detail.

Line 354-356 – this could provide a critical framing of the paper, part of the rationale for the B-E-H idea in the first place.

Response: Good point - we included this idea in the second paragraph of the paper, now line 82-85.

Line 366 – Action Item 2 - a key focus here is also to enable biosphere models to link to other kinds of global models. For example, LeClere et al 2020 shows how different kinds of biodiversity models can be used in combination with different kinds of climate impact models (IAMs more generally), giving better insights about multidimensional /multi-driver environmental change

Response: We have acknowledged this paper and other efforts to combine different impact models, while we still note that these examples, including LeClere et al, do not include feedbacks.

Item 3 Observe biodiversity, ecosystem functioning and human activity change together: we are observing these changes together. The issue is as much about scientific disaggregation.

Response: We are not convinced that we are observing these together. The examples with which we are familiar do not have coordinated observations of each component over time. We left this as it is.

New statistical approaches, comprehensive, Kühl et al 2020: There are good intentions in this section that recognise the history of colonialisms and exploitation, but still take care to avoid hubris overall – there is a tone of “when everyone sees the world as we do...”

Response: We have revised the last sentence here to emphasize investment and other agendas, not just the one we propose here.

Item 4: It would be interesting to compare with atmospheric sciences: “observational programs should be guided by theory that includes feedbacks, and coupled with experimental programs to understand feedbacks.”

Response: We agree! We regret that we do not have the expertise, nor the space, to do this effectively in this paper, though.

Item 5 – Identify a leadership team from where?! Under what aegis? There used to be Diversitas. Any lessons from that and similar international networks?

”A leadership team must assemble, must be able to draw on existing scientific knowledge and work with the research community to develop research programs” What would be the consequences instead of networked working, without head honchos “drawing on” other people’s work? What other kinds of social organization could be thought of?

Response: We revised this to be more open ended with regard to the social organization of the leadership group.

Conclusion – it is over 30 years since biodiversity became a thing. But feedbacks of human wellbeing and life have a much longer history.

Response: We revised this [line 549] to acknowledge the much longer history, good suggestion.

To meet the policy challenges of the coming decades – be specific about these earlier in the paper.

Response: We added a sentence on these earlier in the paper, in Section 3.

Line 425 - "We risk making decisions based on modeled futures that do not capture the full range of likely possibilities (21,50,53,65,85,86)." This sentence has lots of refs tacked on at the end – argue a clear case from these studies / opinions earlier in the paper.

Response: These references each make the same point we make in this sentence and we have cited all of them previously in the paper. So we chose to revise this sentence to make it clear why we have these references at this point [line 562].

Other comments:

Glossary: Fix the ecosystem services / NCP tangle. If the contested nature of the concept matters, explain this properly in the paper itself. Ecosystem services as part of the function-service-benefit chain would be coherent with the B-E-H system presented in the paper. These diverse framings are perhaps a central part of the challenge that feedbacks modelling encounters.

Response: We revised the reference to ecosystem services and nature's benefits to people in the first paragraph, smoothing the introduction of the concept and avoiding the differences. The contested nature of the concept is not critical, for our main points in this paper, as noted in the reviewer's comment. It appears that in our last draft that was reviewed, ecosystem services was listed in the glossary twice! That was a mistake, so we have removed one of those entries.

Figure 1 – unclear diagrams, low quality figures. The article is not a picture book – its needs scientific visualizations. The Venn diagram doesn't makes sense. Ecosystem services in the "humans" system, biodiversity and ecosystem function in the "nature" system

Response: We are trying to avoid the construct of a human and a nature system, as we have noted this is one of the problematic aspects of the IPBES framing. We do see that the figures were not clear, so we have revised them to break down the direct, indirect and feedback effects, and to clarify the examples. We have chosen to keep them as diagrams rather than to include data to keep the general message as the focus.

Figure 2 is about closed and open systems, or systems with defined boundaries. It also shows a switch between hypothesis-led science and other framings of inquiry.

Response: We added a reference to the closed system of single population feedbacks in the figure legend. In the interest of word limits, we did not expand further into the other interesting parts of this comment.

Figure 3 – Feedbacks also relate to system scale – degree of connection, nodality

Response: Very true! We had to make cuts to the manuscript to remain within the length limits and we could not find a way to include this point, which would require additional definitions of terms and explanations.

Abstract P5 - formally, feedbacks are a feature not a component

We changed the word here to ‘feature’, as suggested.

P6 heading - Dynamic feedbacks are causes and consequences

We revised this heading as suggested.

Line 68 – defining biodiversity – from genes to social-ecological systems and beyond...

It’s not clear that this is a suggestion, it is just a restatement of our text. We removed the words ‘and beyond’.

Line 70 – functioning, services, NCP – glossary points out these are different worldviews ontologies.

We sidestepped this discussion in this paragraph with a minor revision:

“Biodiversity, its responses to human activities and the benefits it can provide to human well-being are now at the center of global science-policy initiatives...” [line 73].

P7 Line 107 – feedbacks make human and natural systems change in non-intuitive ways? Coupled systems are of interest here. The literature cited is about feedbacks in modelling to improve physical climate / biophysical Earth system representation. CHANS are still outside this system.

We revised this sentence to refer to systems without specifying human systems to be consistent with those references. The point about human systems is made effectively in the next sentence, so our overall argument was not altered by this edit.

Big statement line 126 – explain more simply – feedbacks explain change and stability.

We have changed accordingly: [line 141] Feedbacks explain change and stability in systems involving biodiversity, ecosystem functioning and human well-being.

Line 136 – disease can have positive feedbacks – pandemics are obvious example. First define the system as a diverse system at the start of the paragraph.

We added a note to indicate that density dependent interactions can be involved in positive feedbacks [line 163], and noted that systems are diverse [line 154], as suggested.

P8 line 152 – explain clearer. The simple story is bad ag and land use practices cause lower crop yields and economic losses (value). Adding a biodiversity component to the story explains why just changing practices and replanting the land won't necessarily fix the problem – if pollinators have been driven away / to extinction. THEN it becomes clearer that the role of biodiversity in feedbacks has not been made clear to policy people.

Response: We have revised this example as recommended by reviewer 1, to provide more references and detail about the feedbacks and the emphasis on biodiversity.